# Synergism between IL7R and CXCR4 drives BCR-ABL induced transformation in Philadelphia chromosome-positive acute lymphoblastic leukemia

Hend Abdelrasoul[1], Anila Vadakumchery[1], Markus Werner[1], Lennart Lenk[2], Ahmad Khadour[1], Marc Young[1], Omar El Ayoubi[1], Fotini Vogiatzi[2], Markus Krämer[1], Vera Schmid [1], Zhengshan Chen[3], Yasar Yousafzai[4], Gunnar Cario[2], Martin Schrappe[2], Markus Müschen [3], Christina Halsey [4], Medhanie A. Mulaw [5], Denis M. Schewe [2], Elias Hobeika[1], Ameera Alsadeq [1,6] & Hassan Jumaa [1,6 ✉]

Ph[+] acute lymphoblastic leukemia (ALL) is characterized by the expression of an oncogenic fusion kinase termed BCR-ABL1. Here, we show that interleukin 7 receptor (IL7R) interacts with the chemokine receptor CXCR4 to recruit BCR-ABL1 and JAK kinases in close proximity. Treatment with BCR-ABL1 kinase inhibitors results in elevated expression of IL7R which enables the survival of transformed cells when IL7 was added together with the kinase inhibitors. Importantly, treatment with anti-IL7R antibodies prevents leukemia development in xenotransplantation models using patient-derived Ph[+] ALL cells. Our results suggest that the association between IL7R and CXCR4 serves as molecular platform for BCR-ABL1-induced transformation and development of Ph[+] ALL. Targeting this platform with anti-IL7R antibody eliminates Ph[+] ALL cells including those with resistance to commonly used ABL1 kinase inhibitors. Thus, anti-IL7R antibodies may provide alternative treatment options for ALL in general and may suppress incurable drug-resistant leukemia forms.

[1] Institute of Immunology, Ulm University Medical Center, 89081 Ulm, Germany. [2] Department of Pediatrics I, ALL-BFM Study Group, Christian-Albrechts University Kiel and University Medical Center Schleswig-Holstein, Kiel, Germany. [3] Department of Systems Biology and City of Hope Comprehensive Cancer Center, Monrovia, CA, USA. [4] Institute of Cancer Sciences, College of Medical, Veterinary and Life Sciences, University of Glasgow, Glasgow, UK. [5] Institute of Experimental Cancer Research, Medical Faculty, University of Ulm, Ulm, Germany. [6] These authors contributed equally: Ameera Alsadeq, Hassan Jumaa. ✉email: hassan.jumaa@uni-ulm.de

The Philadelphia chromosome (Ph) is the most frequent abnormality among adults with acute lymphoblastic leukemia (ALL) (25–30%) and results in *BCR-ABL1* fusion gene[1]. Furthermore, 3–5% of children harbor this translocation which is associated with a poor prognosis[2,3]. As this oncogene confers constitutive kinase activity, addition of tyrosine kinase inhibitors (TKIs) such as imatinib mesylate to intensive chemotherapy has improved the outcome of BCR-ABL1-positive leukemia to a 5-year disease-free survival rate in children (70 ± 12%, $n = 28$)[3]. Nevertheless, Ph[+] ALL patients still suffer from poor prognosis in both children and adults as relapse frequently occurs after stem cell transplantation. A deep understanding of the molecular mechanisms which are associated with BCR-ABL1 transformation is of high importance in order to provide better treatment for these patients and to overcome TKI-resistance. Recently, our group has shown that interleukin 7 receptor (IL7R) is widely expressed in B cell precursor-ALL (BCP-ALL), and that high expression levels of IL7R are correlated with central nervous system involvement (CNS) and may predict CNS-relapse[4].

The cytokine IL7 binds to IL7Rα chain that hetero-dimerizes with the common gamma chain (γc) to form the IL7 receptor and induces the kinase activity of JAK1/JAK3[5]. Alternatively, the IL7Rα chain hetero-dimerizes with the cytokine receptor-like factor 2 (CRLF2) to form the thymic stromal lymphopoietin receptor and mediate activation of JAK1/JAK2[6]. The constitutive expression of IL7R[7,8] in ALL together with the high frequency of mutations affecting IL7R signaling point to a key role of IL7R in disease pathogenesis[9–11]. Thus, investigating the regulation of IL7R function is important for understanding its role in the pathogenesis of ALL. Moreover, characterizing the molecular interaction of IL7R might provide crucial insights into the mechanisms of malignant transformation.

Available data suggest that IL7R expression is controlled by the Forkhead box transcription factor 1 (FOXO1) in lymphocytes[12]. Importantly, FOXO1 is essential during early B cell development and its activity is negatively regulated by phosphatidylinositol-3-kinase (PI3K) signaling[13]. Therefore, FOXO1 function depends on the lipid phosphatase PTEN (phosphatase and tensin homolog) which counteracts PI3K function[14].

The C-X-C chemokine receptor 4 (CXCR4) is a G-protein-coupled receptor which is widely expressed on hematopoietic stem cells and hematopoietic cancers. Together with its ligand CXCL12 (also known as stromal-derived factor 1), CXCR4 plays an important role in tumorigenesis by regulating survival, migration, homing, and interaction of leukemia cells with their microenvironment[15]. High CXCR4 protein expression is correlated with an increased risk of relapse and poor outcome in pediatric ALL patients[16]. Interestingly, CXCL12 was initially identified as a soluble factor that collaborates with IL7 to activate the proliferation of progenitor B cells[17,18].

In this study we have investigated the molecular mechanisms, which are regulated by the oncogenic kinase BCR-ABL1 and are required for malignant transformation or for rescue from kinase inhibitor treatment. We show that IL7R and CXCR4 interact on the cell surface and that both are crucial for malignant transformation of early B cells by BCR-ABL1. Importantly, we show that anti-IL7R antibody can efficiently eliminate inhibitor-resistant Ph[+] patient ALL in preclinical xenograft model.

## Results

### BCR-ABL1 alters IL7R and CXCR4 regulated genes expression.
To better understand the molecular mechanisms regulating BCR-ABL1-induced transformation and the development of Ph[+] ALL, we performed RNA-sequencing (RNA-Seq) and compared transcriptome profile of transformed cells with wildtype (WT) pre-B cells. To this end, six individually generated control WT pre-B cell lines and six BCR-ABL1-transformed pre-B cell counterparts were analyzed. Global transcription profile based principal component analysis (PCA) showed clear segregation of WT and BCR-ABL1-transformed cells (cumulative explained variance = 86.1%; Supplementary Fig. 1a). Gene Ontology (GO) analysis for biological processes was performed and genes which were differentially regulated between the two groups were further investigated, particularly genes related to lymphocyte activation, proliferation, and migration (Fig. 1a and Supplementary Data 1). The analysis showed a differential regulation in multiple signaling pathways related to IL7R signaling (Fig. 1a and Supplementary Fig. 1b). To assess the importance of IL7R signaling related pathways and processes in BCR-ABL1, we performed Gene Set Enrichment Analysis (GSEA) on IL7R related KEGG and REACTOME MSigDB gene sets (Broad Institute, Inc., Massachusetts Institute of Technology (MIT), and Regents of the University of California). Of the eight gene sets analyzed, five showed statistically significant upregulation in BCR-ABL1 as compared with control samples (false discovery rate (FDR) < 0.25; Fig. 1b, Supplementary Fig. 1b, and Supplementary Table 1). Interestingly, genes involved in JAK/STAT signaling, signaling by interleukins, and cytokine signaling were among the most significantly altered gene sets (Fig. 1b and Supplementary Fig. 1b). CXCR4 pathway, though not statistically significant, showed positive correlation to the BCR-ABL1-transformed samples (Supplementary Fig. 2). In addition, the expression of several cytokine signaling regulators such as the transcription repressor *Bcl6* as well as several phosphatases including *Ptpn6*, *Ptpn22*, and *Dusp10* were deregulated by BCR-ABL1 (Fig. 1a). In this work, we focused on the role of IL7R and CXCR4.

### IL7 rescues BCR-ABL1[+] cells from inhibitor treatment.
Our data suggest that the signaling pathways of IL7R and CXCR4 are tightly regulated by the activity of the oncogenic kinase BCR-ABL1 and therefore we hypothesized that they might be directly involved in malignant transformation. To test whether the expression of IL7R and CXCR4 is also correlated in primary ALL, we analyzed a cohort of 68 Ph[+] BCP-ALL patients (patients' characteristics are given in Supplementary Table 2) and found significant correlation of *IL7R* and *CXCR4* gene expression (Spearman $r = 0.6264$; $p < 0.0001$; Fig. 2a), suggesting that our sequence analysis of BCR-ABL1-transformed pre-B cells is in accordance with in vivo condition. In addition, searching in a mixed leukemia gene expression study[19] using the R2 database (http://r2.amc.nl) showed that *IL7R* and *CXCR4* are expressed at reduced levels in BCR-ABL[+] ALL (t9; 22) in comparison with other BCP-ALL entities (Supplementary Fig. 3a and Supplementary Table 3). Similar results were also observed in RNA-seq dataset of 1223 BCP-ALL patients[20] (Supplementary Fig. 3b and Supplementary Table 3).

Interestingly, the inhibition of BCR-ABL1 kinase by imatinib treatment resulted in an upregulated expression of both the chemokine receptor CXCR4 and the IL7R together with downstream signaling elements such as *Jak1* and *Stat5a* (Fig. 2b, c, and Supplementary Fig. 4a). To test whether the upregulation of IL7R or CXCR4 expression upon BCR-ABL1 kinase inhibition affects the survival of BCR-ABL1-transformed cells, we investigated imatinib-induced cell death in the presence of the respective cytokine/chemokine. We found that treatment with IL7 counteracted imatinib-induced cell death and restored the cell cycle progression (Fig. 2d and Supplementary Fig. 4b). However, treatment with CXCL12, the ligand for CXCR4, or its antagonist AMD3100 did not affect imatinib treatment (Fig. 2e and Supplementary Fig. 4b, c). Likewise, treatment with TSLP, the

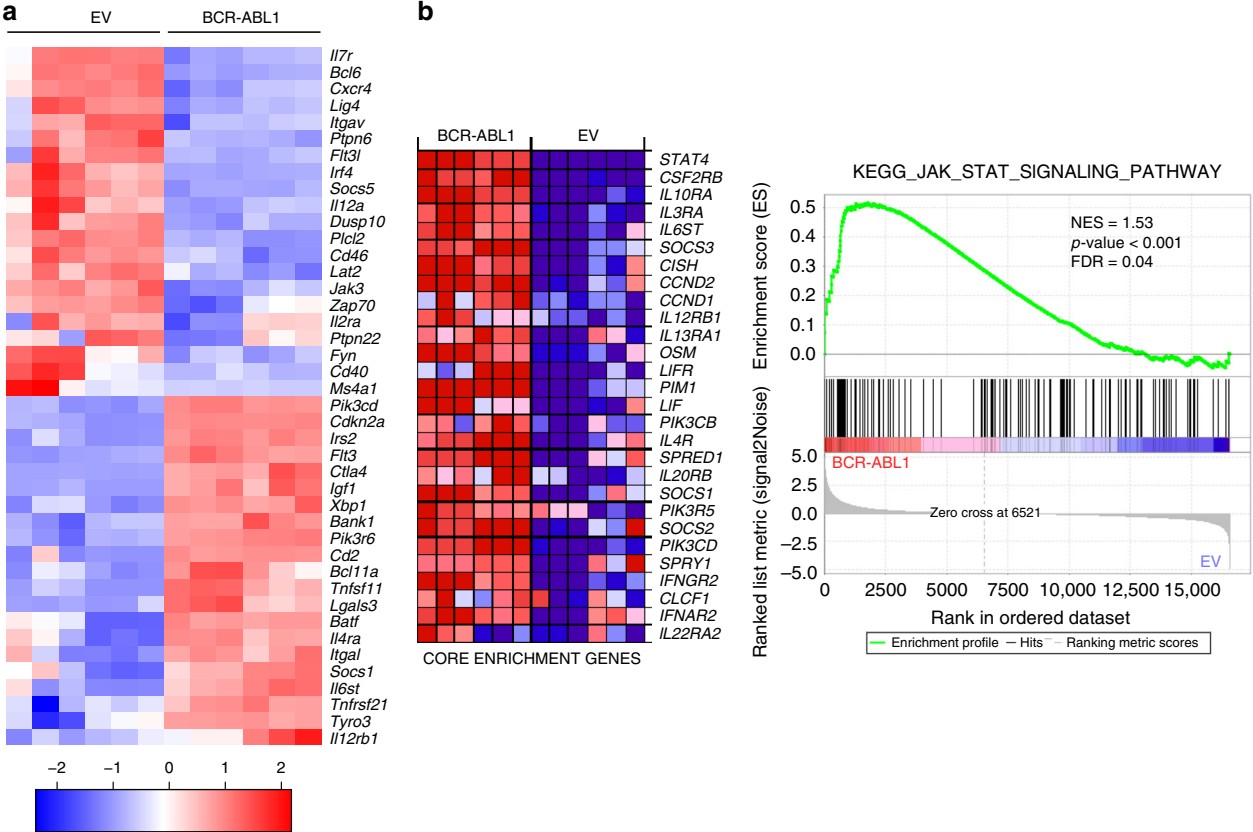

**Fig. 1 Gene expression profiles of BCR-ABL1-transformed cells.** Bone marrow (BM)-derived pre-B cells isolated from two WT mice were used to generate six independent BCR-ABL1-transformed cell lines or six control cell lines expressing empty vector (EV). For expression profiling, RNA was isolated using ReliaPrep™ RNA Miniprep System (MM). All samples were subjected to RNA quality control test before the RNA-seq was applied. **a** Heatmap representation of selected genes related to cytokine receptor signaling from the previously specified GOs (see Supplementary Data 1) in BCR-ABL1-transformed cells compared with the EV-transduced cells. Samples are represented in columns while rows show genes. An average linking method based on Pearson correlation distance metric was applied to cluster rows and columns. **b** Gene Set Enrichment Analysis (GSEA) showing upregulation of gene set belonging to the JAK-STAT signaling pathway in BCR-ABL1 group. Heatmap representation (left) of the top 28 deregulated genes (core enrichment genes) in BCR-ABL1 versus EV-transduced samples (Blue: downregulated, Red: upregulated, NES normalized enrichment score, FDR false discovery rate). A two-sided signal-to-noise metric was used to rank the genes. For a calculated GSEA nominal *p* values of 0, we present them as *p* < 0.001 (otherwise, exact *p* values are shown). Multiple hypothesis testing correction is represented by the estimated FDR.

ligand for CRLF2, was also unable to rescue BCR-ABL1-transformed cells from inhibitor-induced cell death (Supplementary Fig. 4c). Interestingly, human Ph[+] ALL SUP-15 cells also upregulated IL7R and CXCR4 in response to imatinib treatment (Supplementary Fig. 4d). Together, these data suggest that Ph[+] ALL cells upregulate growth factor receptors including IL7R which might enable the survival of Ph[+] cells in microenvironments containing IL7 despite ABL1 kinase inhibitor treatment.

**BCR-ABL1 transformation requires IL7R expression**. The upregulation of IL7R under imatinib treatment raised the question whether IL7R expression is required for BCR-ABL1 induced pre-B cell transformation and ALL development. Therefore, we generated BCR-ABL1-transformed BM-derived pre-B cells from mice homozygous for *loxP*-flanked *Il7rα* alleles (*Il7rα^(fl/fl)*)[21]. Usually, pre-B cells proliferate in the presence of growth factors such as IL7. However, the expression of BCR-ABL1 results in growth factor-independent proliferation in the absence of IL7 (Fig. 3a, b). For inducible deletion of the *Il7rα gene*, we introduced a tamoxifen (Tam)-inducible Cre (Cre-ER^T2) into the BCR-ABL1-transformed *Il7rα^(fl/fl)* cells. Inducible deletion of the *Il7rα* gene led to cell death of the BCR-ABL1-transformed pre-B cells (Fig. 3c–e). To determine the role of IL7R expression in vivo, we injected BCR-ABL1-transformed *Il7rα^(fl/fl)* pre-B cells into

NOD-SCID immunodeficient recipient mice. *Il7r* deletion by Tam treatment in vivo reduced leukemic cell burden and significantly prolonged the survival of xenograft mice injected with BCR-ABL1-transformed cells (Fig. 3f, g). In support of these results, BM-derived cells from *Il7rα*-deficient mice[22] did not give rise to BCR-ABL1-transformed pre-B cells, while BCR-ABL1-transformed myeloid cells (CD11b[+]) can readily be generated from the same cells (Supplementary Fig. 5). Together, our data suggest that IL7R expression is specifically required for the initiation and the maintenance of pre-B cell transformation and ALL development.

**IL7R synergizes with CXCR4 for BCR-ABL1[+] transformation**. Since CXCR4 regulated genes were also altered, we tested whether CXCR4 is also required for BCR-ABL1-induced transformation. Therefore, we generated BCR-ABL1-transformed pre-B cells from mice homozygous for *Cxcr4 loxP*-flanked alleles[23] (*Cxcr4^(fl/fl)*). Deleting *Cxcr4* in these cells using Cre-ER^T2 resulted in rapid cell death and inability of BCR-ABL1 cells to form colonies in vitro (Supplementary Fig. 6).

Since BCR-ABL1 was reported to be involved in crosstalk with CXCR4[24], we investigated whether the requirement for IL7R and CXCR4 in BCR-ABL1-induced transformation is mediated by spatial receptor colocalization. We first examined the effect of

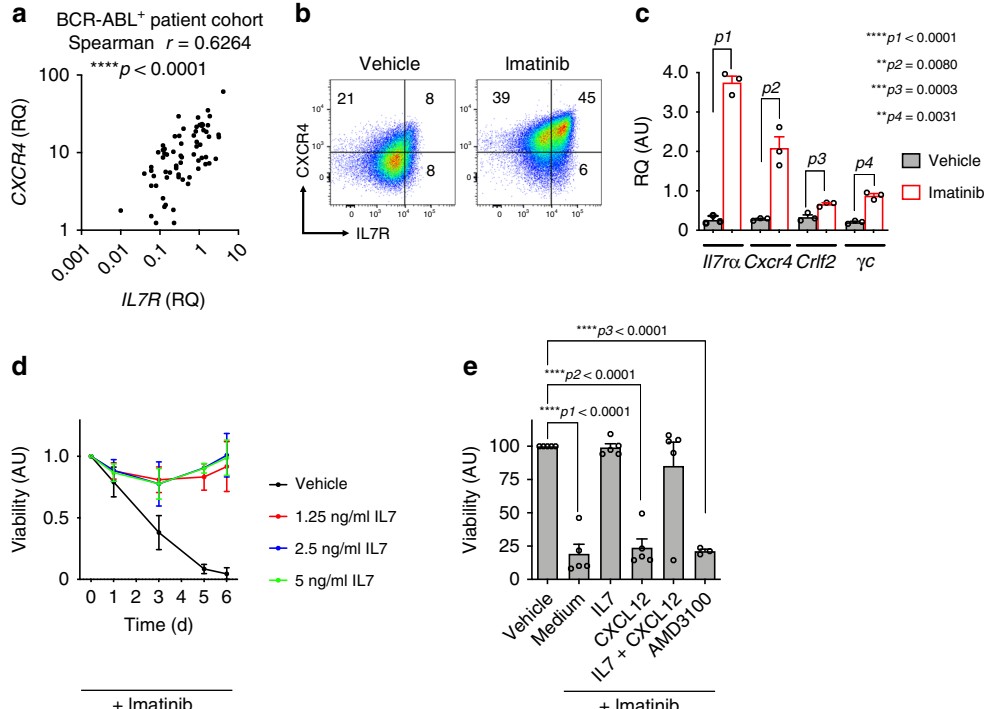

**Fig. 2 Regulation of IL7R and CXCR4 expression levels in BCR-ABL+ ALL. a** Correlation analysis between *IL7R* and *CXCR4* expression levels in 68 pediatric BCR-ABL+ ALL patients. Two-tailed Spearman correlation analysis; exact *p* value = 0.000000011054191. **b** Flow cytometry analysis showing that imatinib treatment (1 μM; 15 h) leads to increased expression of IL7R and CXCR4 on the cell surface of BCR-ABL1-transformed cells. The results are representative of three independent experiments. **c** Quantitative RT-PCR showing that *Il7r* and *Cxcr4* and other associated factors are regulated at the level of transcription. *N* = 3 independent samples per group, and error bars represent mean ± SEM. Unpaired *t*-test, two-sided, p1 exact value = 0.000035569652955. RQ: relative quantification; AU arbitrary unit. **d** BCR-ABL1-transformed WT pre-B cells were treated with 1 μM imatinib and with different concentrations of IL7 as indicated for 6 days. *N* = 3 independent samples per group, and error bars represent mean ± SD. One-way ANAOVA. Dunnett's multiple comparisons test was performed to day 6, compared with control group (Vehicle). Adjusted *p* values: Vehicle vs. 1.25 ng/ml IL7 ***p = 0.0003, Vehicle vs. 2.5 ng/ml IL7 ***p = 0.0002, Vehicle vs. 5 ng/ml IL7 ***p = 0.0002. **e** Treatment of BCR-ABL1-transformed cells with imatinib leads to cell death and concomitant incubation with IL7, but not CXCL12, reverses this effect. *N* = 3 independent samples per group, and error bars represent mean ± SEM. Unpaired *t*-test, two-sided, ****p1 exact value = 0.000003416387742, ****p2 exact value = 0.000002948485097, ****p3 exact value = 0.000000000342967.

BCR-ABL1 recruitment on CXCR4-mediated $Ca^{2+}$ mobilization[25]. Therefore, the CXCL12-induced $Ca^{2+}$ flux in WT as compared with BCR-ABL1-transformed cells was tested. While WT cells showed a negligible CXCL12-induced $Ca^{2+}$ flux, BCR-ABL1-transformed cells showed a robust $Ca^{2+}$ response (Fig. 4a). Inhibiting the BCR-ABL1 kinase activity by either imatinib or dasatinib blocked the CXCL12-induced $Ca^{2+}$ response with dasatinib showing an effective inhibition at much lower concentrations than imatinib, which is most likely caused by the additional effect of dasatinib on Src kinases[26] (Fig. 4a). Importantly, inducible deletion of *Cxcr4* in BCR-ABL1-transformed cells or treating them with AMD3100, an antagonist of CXCL12, prevented the $Ca^{2+}$ response (Supplementary Fig. 7a). These data are in full agreement with the view that BCR-ABL1 is recruited to CXCR4 and can be activated by the respective ligand CXCL12.

Interestingly, the *Cxcr4*-deficient pre-B cells showed an increased differentiation capacity as measured by the elevated ratio of cells expressing the immunoglobulin kappa light chain (Supplementary Fig. 7b). These data suggest that CXCR4 cooperates with IL7R in preventing pre-B cell differentiation[27]. Similarly, IL7R seems to act together with CXCR4 in directing cell migration, as both *Cxcr4*-defcient and *Il7r*-defcient BCR-ABL1-transformed cells show an impaired migration towards a CXCL12 gradient (Supplementary Fig. 7c).

To study further how IL7R and CXCR4 act synergistically to regulate pre-B cell differentiation and proliferation, we

investigated the interaction between IL7R and CXCR4 by proximity ligation assay (PLA). Adjacent binding of the ligands IL7 (7 kD) and CXCL12 (15 kD) suggests that the corresponding receptors are localized on the cell surface at a proximity below 10 nm in precursor B cells (Fig. 4b and Supplementary Fig. 8a). Interestingly, BCR-ABL1-transformed pre-B cells show an increased number of IL7R/CXCR4 foci as compared with untransformed WT pre-B cells (Fig. 4c and Supplementary Fig. 8b). This association is also detected in human BCR-ABL+ pre-B ALL cells (Supplementary Fig. 8c). The interaction between CXCR4/IL7R and BCR-ABL was further confirmed by immunoprecipitation (Supplementary Fig. 8d). Together, these findings suggest that the interaction between IL7R and CXCR4 is increased in BCR-ABL1-transformed pre-B cells. Hence, we postulated that this interaction recruits IL7R-associated signaling proteins into close proximity to CXCR4 thereby enabling activation by BCR-ABL1 which then leads to pre-B cell transformation. To directly test this hypothesis, we investigated the association of JAK3 with CXCR4 in BCR-ABL1-transformed cells as compared with WT control. Usually, JAK3 is associated with the γc subunit of IL7R but not with CXCR4. In fact, CXCL12/CXCR4 signaling was shown to be independent of JAK3[28]. Interestingly, a significant IL7R-dependent increase in JAK3 association with CXCR4 was observed in BCR-ABL1-transformed pre-B cells (Fig. 4d). Similarly, an IL7R-dependent association of CXCR4 with pJAK3 was observed in BCR-ABL1-transformed cells (Supplementary Fig. 8e). In contrast, the

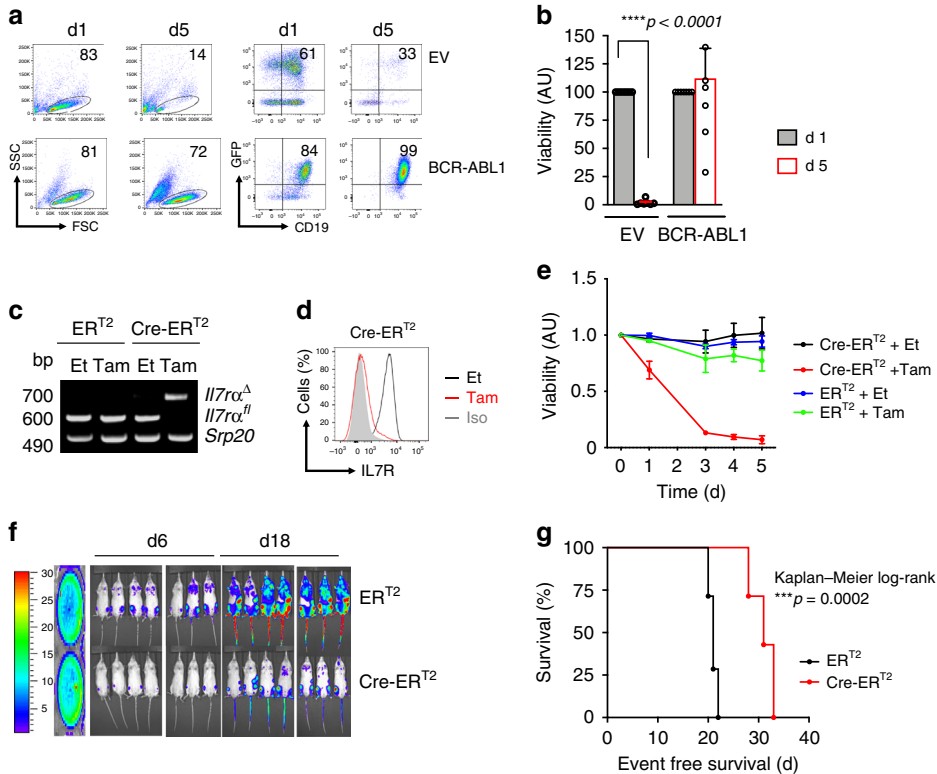

**Fig. 3 IL7R is required for BCP-ALL leukemogenesis. a** BM-derived pre-B cells from WT mice were transduced with either an empty vector (EV) or with BCR-ABL1. Viability of the cells after IL7 withdrawal was determined by flow cytometry (left). The enrichment of CD19+GFP+ (right). The results are representative of three independent experiments. **b** Quantification of viable CD19+GFP+ cells relative to control after IL7 withdrawal ($n = 7$ per group, and error bars represent mean ± SEM. Unpaired $t$-test, two-sided, exact $p$ value ****$p < 0.000000000000001$). AU arbitrary unit. **c–g** $Il7ra^{fl/fl}$ pre-B cells were transformed with BCR-ABL1 and were then transduced with either ER$^{T2}$ or Cre-ER$^{T2}$. Cells were treated with either tamoxifen (Tam) to induce Cre expression, or with ethanol (Et). **c** PCR analysis for $Il7ra$ deletion ($Il7ra^{\Delta}$), $Srp20$ was used as a loading control. Cells transduced with ER$^{T2}$ were used as control. The results are representative of three independent experiments. bp base pair. **d** Extracellular staining for IL7R protein in $Il7ra^{fl/fl}$ pre-B cells after 72 h of Cre-induction. The results are representative of three independent experiments. **e** The percentage of living cells were determined by flow cytometry using Sytox as an excluding dead cell stain. $N = 3$ independent samples per group, and error bars represent mean ± SD. One-way ANAOVA. Dunnett's multiple comparisons test was performed to day 5, compared with control group (ER$^{T2}$ Et). Adjusted $p$ values: ER$^{T2}$ Et vs. ER$^{T2}$ Tam n.s. $p = 0.1769$, ER$^{T2}$ Et vs. Cre-ER$^{T2}$ Et n.s. $p = 0.6325$, ER$^{T2}$ Et vs. Cre-ER$^{T2}$ Tam ****$p < 0.0001$. **f** Luciferase bioimaging and **g** survival curves of NOD-SCID mice that were injected with $1 \times 10^6$ $Il7ra^{fl/fl}$-BCR-ABL1 pre-B cells containing either a Cre-ER$^{T2}$ construct or an ER$^{T2}$. ($n = 21$ per group). Mantel-Cox-log-rank test, ***$p$ value $= 0.0002$.

association between IL7R and JAK3 showed no significant change suggesting that BCR-ABL1-induced transformation has no effect on IL7R interaction with its downstream signaling elements (Supplementary Fig. 8f).

As expected, JAK kinases show increased phosphorylation in BCR-ABL1-transformed cells and imatinib treatment reduces this phosphorylation (Fig. 4e). In full agreement with the hypothesis that the interaction between CXCR4 and IL7R enables CXCR4 to utilize the downstream signaling machinery of IL7R, inducible inactivation of either IL7R or CXCR4 expression results in decreased activity of JAK1, JAK2, and JAK3 as shown by their reduced phosphorylation (Fig. 4e).

Together, these data suggest that BCR-ABL1 interaction with CXCR4 recruits this oncogene into the proximity of IL7R-associated JAK kinases thereby enabling their BCR-ABL1-mediated activation and pre-B cell transformation.

**BCR-ABL1 controls IL7R expression by regulating FOXO1.** Activated JAK kinases phosphorylate the cytoplasmic domain of cytokine receptors at specific tyrosine residues leading to the recruitment and subsequent activation of signal transducer and activator of transcription (STAT) proteins. Phosphorylated STATs undergo dimerization and translocate to the nucleus

where they activate target genes involved in proliferation and survival of lymphocytes[29]. As expected, increased STAT5 phosphorylation was detected in BCR-ABL1-transformed cells and ABL1 kinase activity was required for this increase (Fig. 5a). Since STAT5 is activated by IL7R signaling[30], and also by BCR-ABL1, we postulated that activated STAT5 might control IL7R expression in a negative feedback loop that prevents deregulated IL7R expression. The transcription factor FOXO1 was shown to regulate the expression of IL7R[12] as well as CXCR4[31,32]. The fact that STAT5 activates PI3K signaling[33] which in turn suppresses FOXO1 transcriptional activity by phosphorylation of specific S/T sites, suggests that STAT5 activation can lead to increased FOXO1 phosphorylation and subsequent downregulation of IL7Rα expression. Indeed, imatinib treatment of BCR-ABL1-transformed pre-B cells resulted in decreased FOXO1 phosphorylation (Fig. 5a). Moreover, introducing a constitutively active STAT5 version (STAT5-CA)[34] into pre-B cells resulted in FOXO1 inactivation, as measured by increased S256 phosphorylation, and decreased IL7Rα expression (Fig. 5b, c). In full agreement of transcriptional repression, reverse transcriptase PCR experiments revealed that $Il7ra$ transcripts were almost missing in STAT5-CA expressing cells (Fig. 5d). Interestingly, the reduced IL7R expression was associated with loss of the cells expressing STAT5-CA (Fig. 5e). These findings suggest that

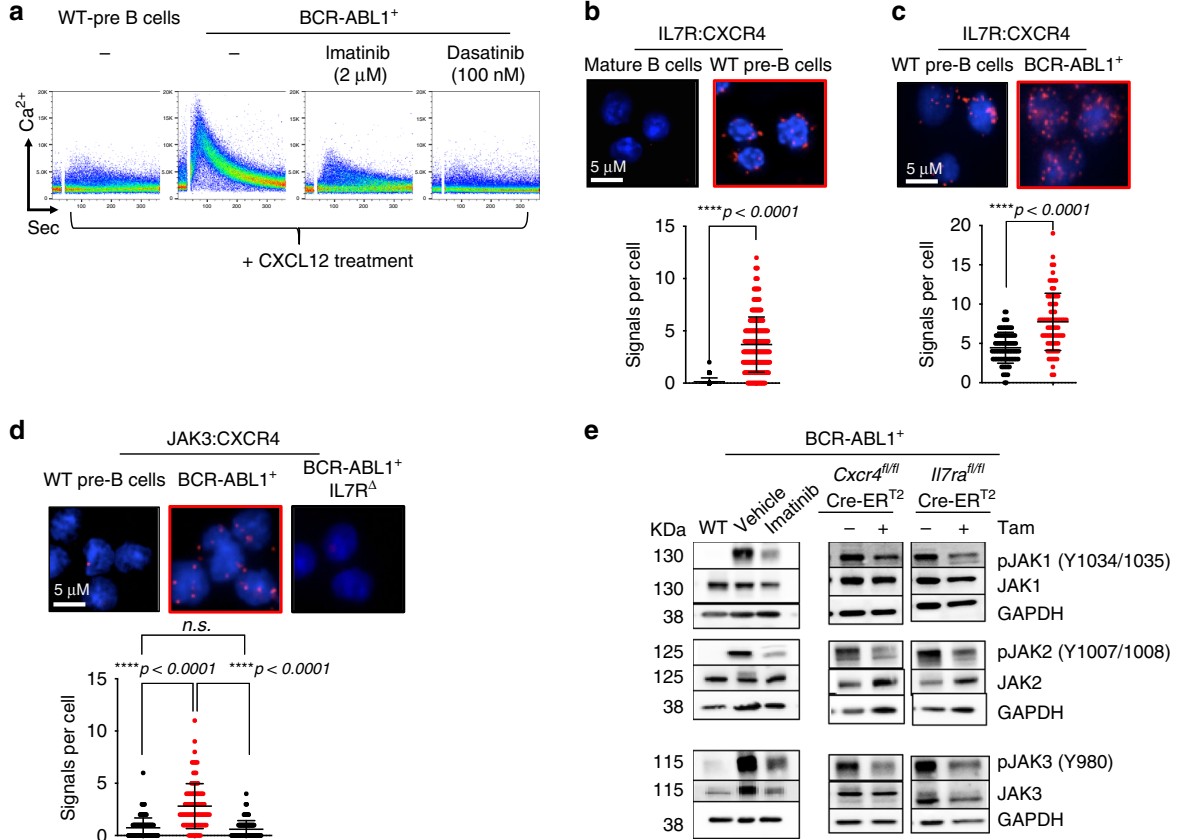

**Fig. 4 Synergism between CXCR4 and IL7R in BCR-ABL1-induced cell. a** BCR-ABL1 acts downstream of CXCR4. Shown are flow cytometry plots for $Ca^{2+}$ mobilization in response to CXCL12 treatment (100 ng/ml) in WT as compared with BCR-ABL1-transformed cells with or without kinase inhibitor treatment as indicated. The results are representative of three independent experiments. **b** Proximity ligation analysis (PLA) to detect the close association of IL7R and CXCR4 on the surface of WT BM-derived pre-B cells as compared with mature B cells (which lacks IL7R expression; Supplementary Fig. 8a). **c** PLAs showing increased association of IL7R and CXCR4 on the surface of BCR-ABL1-transformed cells. **d** PLA showing recruitment of JAK3 to CXCR4 in BCR-ABL1-transformed cells. This association was lost upon inducible deletion of *Il7r* using Cre-ER$^{T2}$ (IL7R$^{\Delta}$) system. Close proximity is represented by red dots. **b–d** Quantification shows number of signals per cell, error bars represent mean ± SD. Unpaired *t*-test, two-sided. The results shown are representative of three independent experiments. Exact *p* values (**c**) 0.00000000006167, approximate *p* value (**b**) <0,000000000000001, approximate *p* value (**d**) <0,000000000000001. **e** Left panel, western blot analysis for increased phosphorylation of the JAK kinases by BCR-ABL1 activity. Right panel, western blot showing that inducible deletion of CXCR4 or IL7R results in reduced phosphorylation of all JAK kinases. The results are representative of three independent experiments. KDa kilo Dalton.

STAT5 regulates IL7R expression in a negative feed-backloop and that fine-tuned STAT5 activity in transformed cells is important to induce cell proliferation and, at the same time, avoid destruction of IL7R expression by excessive STAT5 activity. Altogether, we propose that BCR-ABL1 controls IL7R expression by activating a common STAT5-regulated negative feedback mechanism and that the observed downregulation of IL7R expression by BCR-ABL1 guarantees a fine-tuned STAT5 activity.

**FOXO1 is required for leukemogenesis.** To further confirm the requirement of FOXO1 transcription factor for BCR-ABL1-mediated leukemogenesis, we generated BCR-ABL1-transformed pre-B cells from mice homozygous for *loxP*-flanked alleles of *FoxO1* (*FoxO1*$^{fl/fl}$). To induce *FoxO1* deletion, we introduced into the BCR-ABL1-transformed cells our Tam-inducible Cre-ER$^{T2}$ by retroviral transduction. Inducible deletion of *FoxO1* led to cell loss of the BCR-ABL1-transformed cells (Fig. 6a–c). Moreover, deletion of *FoxO1* resulted in concomitant downregulation of FOXO1 and IL7R expression (Fig. 6d). To provide further evidence for the important role of FOXO1 in leukemogenesis in vivo, we injected BCR-ABL1-transformed *FoxO1*$^{fl/fl}$ cells into

sublethally irradiated NOD/SCID recipient mice and monitored development of leukemia after deletion of *FoxO1* as compared with controls in vivo. We found that BCR-ABL1-transformed *FoxO1*$^{fl/fl}$ caused fatal leukemia within 2 weeks, while deleting *FoxO1* by Tam-induced Cre-ER$^{T2}$ activation reduced the leukemic cell burden and prolonged the survival time of respective mice (Fig. 6e, f).

Together, these data suggest that FoxO1 plays an essential role in BCR-ABL1-induced transformation most likely through the activation of IL7R expression (Supplementary Fig. 9).

**Targeting IL7R prevents BCR-ABL1$^{+}$ leukemia development.** Since the above results show that IL7R is crucial for the transforming signals initiated by BCR-ABL1 in Ph$^{+}$ ALL, we investigated whether inhibition of IL7R signaling using ruxolitinib, a JAK1/JAK2 kinase inhibitor, can interfere with the survival of BCR-ABL1-transformed cells or enhance the effect of kinase inhibitors on these cells. We found that treatment with ruxolitinib along with imatinib prevented the IL7-driven rescue of BCR-ABL1-transformed pre-B cells in vitro (Supplementary Fig. 10a–c). To further study the consequences of ruxolitinib on

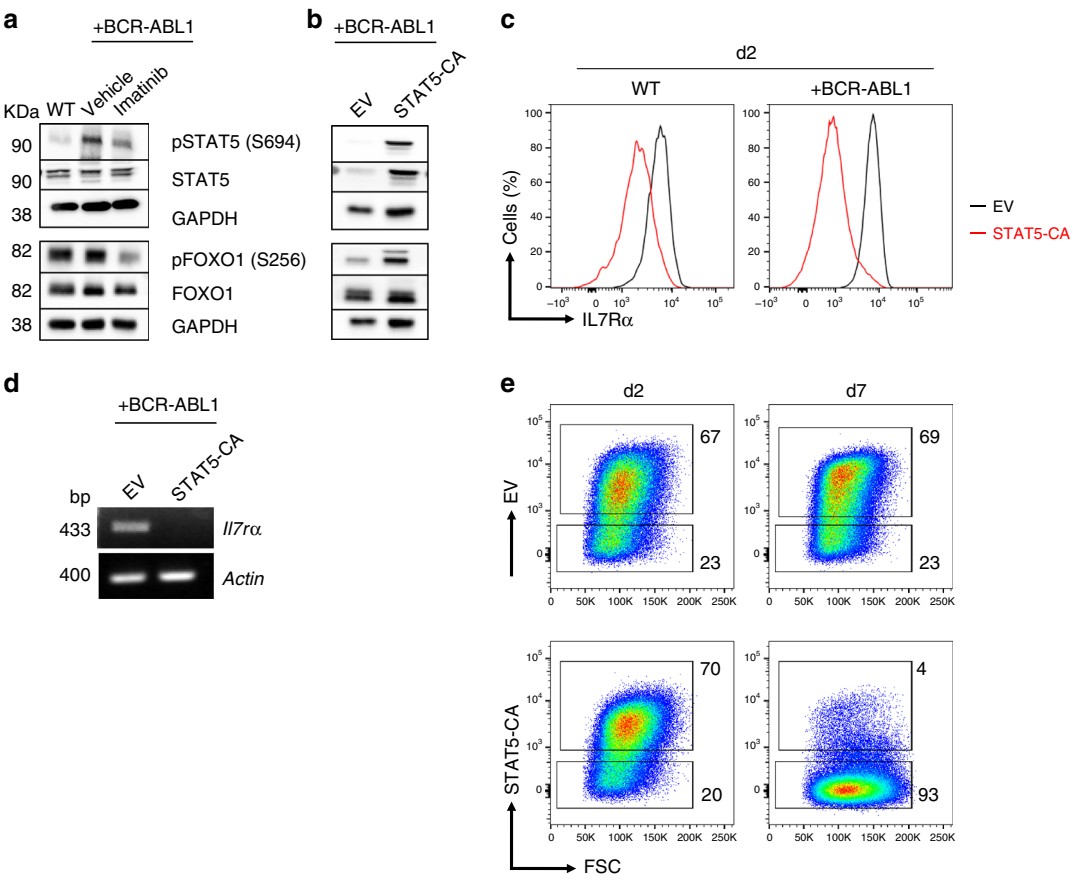

**Fig. 5 Negative feedback regulating BCR-ABL1-induced transformation. a** Western blot analysis for increased phosphorylation of STAT5 and FOXO1 by BCR-ABL1 activity. **b** Western blot analysis showing that constitutively active STAT5 (STAT5-CA) enhances FOXO1 phosphorylation. The western blot results in **a** and **b** are representative of three independent experiments; KDa kilo Dalton. **c** Ectopic expression of STAT5-CA in WT pre-B cells (left) or BCR-ABL1-transformed cells (right) leads to downregulation of surface expression of IL7R protein after 2 days of transduction. **d** Reverse transcriptase PCR analysis showing downregulation of IL7R due to constitutive activity of STAT5; actin was used as a loading control. cDNA was prepared from sorted cells after 2 days of transduction. bp base pair. **e** Loss of BCR-ABL1-transformed pre-B cells which overexpress constitutively active STAT5 (STAT5-CA). Representative flow cytometry plots at day 2 and 7 after transduction. EV empty vector. **a–e** The results are representative of three independent experiments.

leukemia cells in vivo, we injected human BCR-ABL[+] ALL cells into NSG mice and monitored the recipient mice under imatinib, ruxolitinib, or combined imatinib/ruxolitinib treatment. We found that combination treatment was unable to prolong the survival of recipient mice or to reduce the percentage of leukemic cells in the BM and in the spleen (Supplementary Fig. 10d–f) suggesting that, in contrast to the in vitro results, ruxolitinib cannot support imatinib in a xenograft model and therefore may not be suitable for ALL treatment in vivo. Therefore, we tested whether direct targeting of the IL7R using specific monoclonal antibodies may interfere with its function in leukemia. To this end, we injected imatinib-resistant BCR-ABL1[+] ALL patient cells[35] into NSG mice and treated them with monoclonal antibody specific for human IL7Rα[4] (Fig. 7b–d). As expected, imatinib was unable to prevent leukemia development and, therefore, the leukemia burden was increased and the survival of imatinib-treated mice was reduced similar to that of control mice (Fig. 7a, b). In contrast, anti-IL7R antibody significantly delayed leukemia onset in vivo and led to a significantly expanded survival time of the respective animals (Fig. 7a, b and Supplementary Fig. 11a, b). Interestingly, the xenograft patient material showed an upregulation in *BCR-ABL1* expression compared with human BCR-ABL1[+] cell lines TOM-1 and SUP-B15 (Fig. 7c) which may explain the imatinib-resistant

phenotype[35]. Indeed, we found that regulated long-term exposure to imatinib can lead to upregulation of *BCR-ABL1* expression and to imatinib-resistance in vitro (Supplementary Fig. 12). The xenograft patient material which was used lacked the BCR-ABL1 gate keeper mutation, known as T315I kinase domain mutation, which leads to resistance against ABL1 inhibitors[36,37]. Therefore, we repeated the experiment using T315I-positive, imatinib resistant, xenograft patient material[38]. Treatment of recipient mice with anti-IL7R antibody significantly delayed leukemia onset and led to significant prolongation of the survival time of the respective animals (Fig. 7d–f and Supplementary Fig. 11c). As NSG mice lack NK cells, we excluded that it led to cell death via antibody-dependent cell-mediated cytotoxicity as previously described by other IL7R antibodies[39]. Similarly, the antibody which we used does not block IL7 binding (Supplementary Fig. 13a), alternatively, it disrupts the scaffold between IL7R and CXCR4 (Supplementary Fig. 13b). In addition, antibody treatment seems to enhance apoptosis as shown by increased cleavage of caspase-8[40] (Supplementary Fig. 13c).

Together, these experiments show that IL7R plays a pivotal role in the survival of ALL and that targeting IL7R via specific antibodies exerts a profound effect on elimination of kinase inhibitor-resistant ALL in vivo.

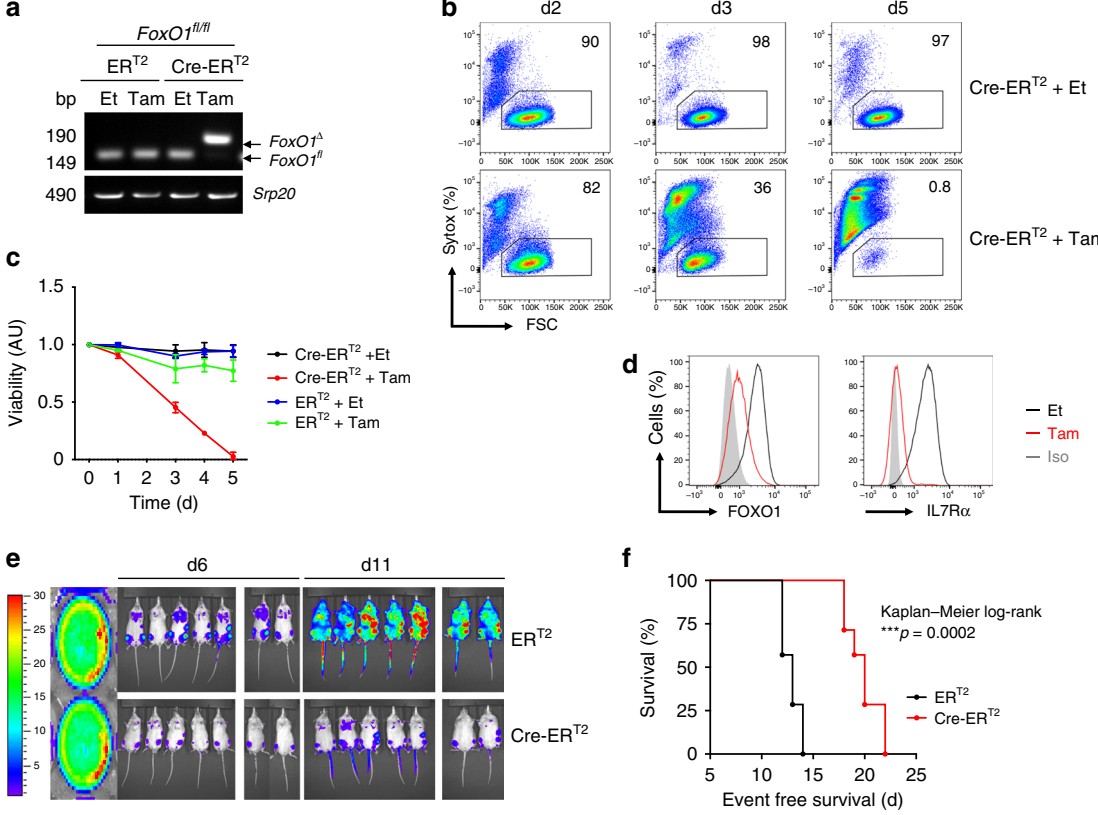

**Fig. 6 *FoxO1* is required for BCR-ABL1 driven leukemogenesis.** *FoxO1$^{fl/fl}$* pre-B cells were transformed with BCR-ABL1 and were then transduced with Cre-ER$^{T2}$. Cells were treated with either tamoxifen (Tam) to induce Cre expression, or with ethanol (Et). **a** PCR analysis for *FoxO1* deletion (*FoxO1$^\Delta$*), *Srp20* was used as a loading control. Cells transduced with ER$^{T2}$ were used as control. bp base pair. **b** The percentage of living cells were determined by flow cytometry using Syotx as excluding dead cell stain. **c** The fold change of living cells after treatment with either Et or Tam at different time points. $N = 3$ independent samples per group, and error bars represent mean ± SD. One-way ANAOVA. Dunnett's multiple comparisons test was performed to day 5, compared with control group (ER$^{T2}$ Et). Adjusted *p* values: ER$^{T2}$ Et vs. ER$^{T2}$ Tam **p* = 0.0251, ER$^{T2}$ Et vs. Cre-ER$^{T2}$ Et n.s. *p* > 0.9999, ER$^{T2}$ Et vs. Cre-ER$^{T2}$ Tam *****p* < 0.0001. AU arbitrary unit. **d** Intracellular staining for FOXO1 protein (left) and an extracellular staining of IL7R (right) after 72 h of Cre-induction. The results are representative of three independent experiments. **e** Luciferase bioimaging and **f** Survival curves of mice that were injected with $1 \times 10^6$ *FoxO1$^{fl/fl}$* cells that had been transformed with BCR-ABL1 and transduced with either a Cre-ER$^{T2}$ construct or an empty vector control (ER$^{T2}$). Cells were labeled with luciferase and injected into NOD-SCID mice. (*n* = 14 per group). Mantel–Cox-log-rank test, ****p* = 0.0002.

## Discussion

Previous studies demonstrated remarkable outcome improvements in Ph$^+$ ALL patients upon imatinib integration into chemotherapy[41]. However, acquired drug resistance is still a crucial issue that leads to relapse of the disease and unfavorable outcome[3,42]. A thorough understanding of the molecular mechanisms involved in BCR-ABL1-mediated transformation is required in order to provide therapeutic alternatives for Ph$^+$ ALL patients, particularly those who developed TKI-resistance. In this study, we employed several genetically modified systems as well as preclinical xenograft models to better understand BCR-ABL1-induced transformation. Interestingly, our data show that BCR-ABL1 oncogene regulates the expression and function of the signaling pathways of IL7R and CXCR4 in a concerted manner. This combined regulation is important because both receptors are required for the growth and survival of BCR-ABL1-transformed pre-B cells. Importantly, IL7R and CXCR4 act in close proximity thereby allowing their downstream signaling pathways to synergize and enable BCR-ABL1-induced pre-B cell transformation. In this synergism, CXCR4 attracts the oncogenic kinase BCR-ABL1 while IL7R conveys the JAK/STAT signaling machinery. Importantly, this complex seems to act in a ligand-independent manner to activate multiple downstream signaling pathways and is

required for the survival of mouse and human leukemia cells in both in vitro as well as in vivo preclinical xenograft model.

Our results indicate that BCR-ABL1 utilizes the IL7R signaling machinery for pre-B cell transformation and growth factor-independent proliferation and that the feedback regulation of this machinery is a crucial part of the transformation process. For instance, deregulated BCR-ABL1 kinase activity may result in uncontrolled STAT5 phosphorylation and negative feedback regulation of IL7R expression leading to cell death. Previous report suggested that BCR-ABL oncogene mimics pre-BCR signaling by activating STAT5 on one hand and repressing BCL6 expression on the other hand[7]. STAT5 was also shown to directly downregulate BCL6 expression in response to IL7 stimulation[43]. This is in agreement with our data showing that BCR-ABL1 transformation downregulates the transcription repressor BCL6. Thus, BCR-ABL1-mediated pre-B cell transformation requires an equilibrium between kinase activity and negative feedback regulation of IL7R signaling. In full agreement, BCR-ABL1-transformed pre-B cells require multiple phosphatases that are most likely involved in stabilizing this equilibrium[38] which can be targeted for efficient treatment of Ph$^+$ ALL. It is feasible that additional players participate in regulating IL7R expression in ALL. For example, it was previously shown that *IKAROS*

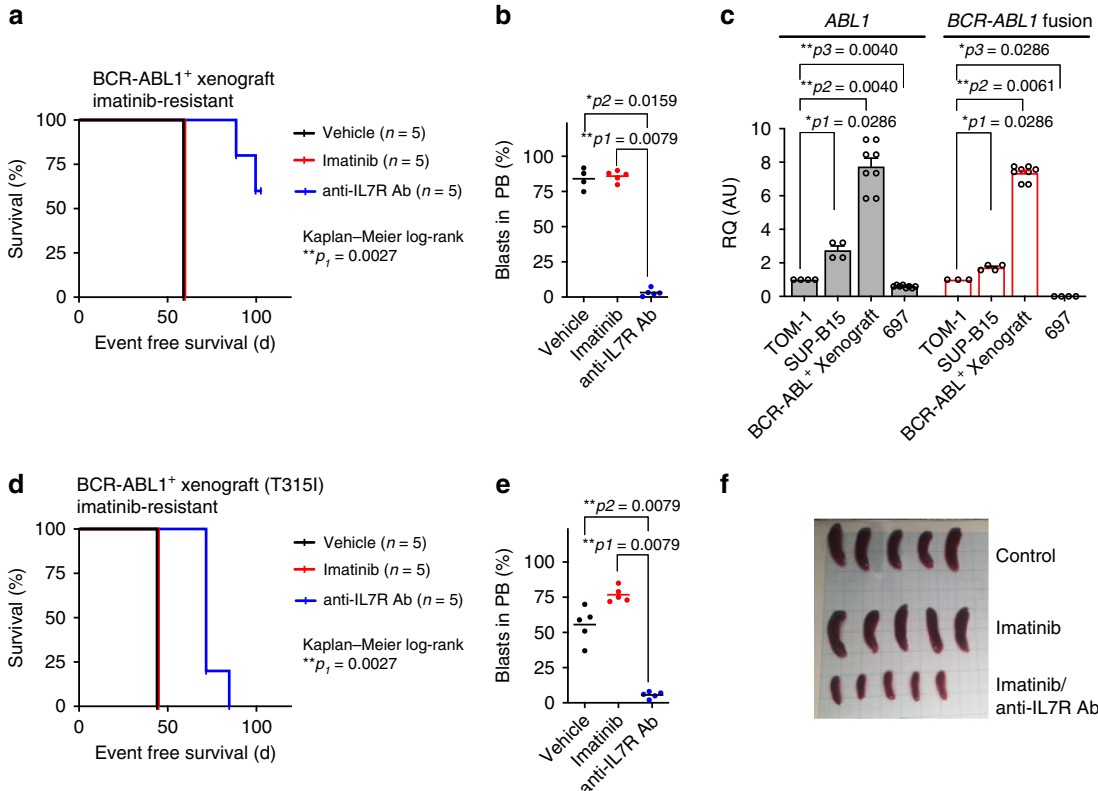

**Fig. 7 Anti-IL7R antibody suppresses BCR-ABL1⁺ ALL development in vivo. a, b** $1 \times 10^6$ BCR-ABL1⁺ ALL patient cells were xenografted by tail vein injection into NSG mice. Xenografted mice were treated with vehicle, imatinib, or with anti-IL7R antibody ($n = 5$ per group) as described in "Methods". **a** Survival prolongation in xenografted mice subjected to the indicated treatment. Mantel–Cox-log-rank test, $**p = 0.0027$. **b** Leukemic engraftment was measured by flow cytometry in peripheral blood (PB) at day 58 and all control and imatinib-treated mice were sacrificed due to appearance of leukemic symptoms. Unpaired t-test, two-sided p value. **c** Expression levels of *ABL1* or *BCR-ABL1* fusion in BCR-ABL1⁺ human cell lines (TOM-1 and SUP-15), and in imatinib-resistant xenograft-derived human BCR-ABL1⁺ cells. 697 cell line, was used as a negative control for the fusion. Expression levels are normalized to TOM-1. Error bars represent mean ± SEM. Unpaired t-test, two-sided. RQ: relative quantification; AU arbitrary unit. **d-f** $1 \times 10^6$ BCR-ABL1⁺ ALL patient cells holding T315I mutation (imatinib-resistant) were injected intravenously into NSG mice and treated with anti-IL7R antibody or with imatinib. **d** Survival prolongation in mice xenografted with Ph⁺ BCP-ALL patient material holding the T315I mutation and treated with either imatinib or with anti-IL7R antibody. Mantel–Cox-log-rank test, $**p = 0.0027$. **e** Leukemic engraftment was measured by flow cytometry in PB at day 42. Unpaired t-test, two-sided. **f** Spleen size at day 45 of xenograft mice treated with either imatinib or a combination of imatinib and anti-IL7R antibody.

negatively regulates *IL7R* promoter and that *IKAROS* deficiency in ALL patients is correlated with increased IL7R expression[44]. Similarly, the common *IKZF1* deletion leading to dominant negative IK6 isoform[45] resulted in increased IL7R expression in BCR-ABL⁺ cells[46]. Thus, co-occurring genomic alterations such as *IKZF1* deletion remain to be addressed in future studies.

Previous reports showed that combined targeting of BCR-ABL1 and JAK2 using dasatinib and ruxolitinib, respectively, reduced leukemia engraftment and prolonged survival[47]. However, these mice eventually relapsed and died from leukemia which suggest that ruxolitinib treatment is inefficient in vivo[47]. This is in agreement with our results showing that inhibition of the kinases JAK1/JAK2 by ruxolitinib, applied either alone or in combination with imatinib, was not able to provide any therapeutic advantage for xenograft animal models injected with Ph⁺ ALL patient material. It is conceivable that reduced drug availability or insufficient inhibition of IL7R signaling, as ruxolitinib mainly inhibits JAK1 and JAK2 while IL7R can also activate JAK3, are responsible for the inability of ruxolitinib to block the development of Ph⁺ ALL in vivo.

Intriguingly, our experiments point to an unpredicted escape mechanism of transformed cells during TKI treatment. Since leukemic cells maintain the expression of growth factor receptors such as IL7R, which is used as scaffold for organizing the oncogenic signaling machinery, the presence of IL7 in certain niches might provide the transformed cells with escape mechanisms upon treatment with inhibitors blocking BCR-ABL1[48]. This scenario is also possible for other growth factor receptors and their respective cytokines. For example, it has been shown that IL3 can rescue BCR-ABL⁺ CML cells from cell death induced by BCR-ABL inhibitors[47,49]. Although our data showed that several receptors were upregulated in response to imatinib (such as IL7R, CXCR4, and CRLF2), IL7 showed a unique potential to rescue the cells under kinase inhibitor treatment. Thus, it is conceivable that, during treatment of Ph⁺ ALL patients with inhibitors blocking BCR-ABL1 kinase activity, IL7R-driven survival pathways in ALL cells are activated in microenvironments containing IL7 thereby enabling the survival of ALL cells. Ph⁺ ALL cells that survive treatment with BCR-ABL inhibitors in microenvironments containing IL7 may act as leukemia initiating cells and disseminate to other locations when inhibitor concentrations decline or when inhibitor resistance is induced by somatic mutations. This scenario is further supported by the elevated amounts of IL7 detected in ALL patients[9,10]. Thus, understanding the molecular mechanisms of BCR-ABL1-induced transformation is important for identifying TKI escape mechanisms and for developing strategies that prevent such escape.

Our findings may also have important therapeutic implications in other leukemia subtypes with similar gene expression such as Ph-like ALL. For example, at least 90% of patients with Ph-like ALL showed kinase-activating alterations (e.g., in *ABL1/ABL2* and *JAK2*), sequence mutations in *IL7R* as well as an activation of phosphorylated STAT5[50]. This suggests that IL7R might also be a potential therapeutic target for several BCP-ALL patients who are not Ph$^+$ as well[4]. Nevertheless, additional work would be required to investigate whether our model also function in Ph-like ALL. Since IL7R expression and function is critical for proper lymphopoiesis, targeting this pathway may have effects on other normal cells. For instance, previous studies showed that mice deficient in *Il7r* showed depletion in both B and T lymphocytes[51]. In humans, mutations in the *IL7Rα* result in severe combined immunodeficiency (SCID) which is associated with the absence of T cells and normal numbers, nevertheless inactive, B cells[52]. Accordingly, targeting IL7Rα using specific antibodies may also affect T cells[53] and lead to immunodeficiency in patients. However, a recent study showed that treating healthy subjects with anti-human IL7R antibody was well tolerated and did not result in obvious alterations in immune cell populations and inflammatory cytokine profiles[54]. Thus, treatment with anti-IL7R antibodies might provide a key therapeutic approach especially for TKI-resistant ALL once the different antibodies are characterized regarding their side-effects and compared with standard chemotherapy in appropriate clinical trials.

## Methods

**Patient samples, human cell lines**. Sixty-eight BCR-ABL$^+$ ALL patients were treated according to European intergroup study of post-induction treatment of Philadelphia chromosome-positive ALL (EsPhALL) 2004 and 2010 protocols (NCT00287105) and ALL-Berlin-Frankfurt-Münster (BFM) 2000 (NCT00430118) study. Informed consent was obtained according to institutional regulations, in accordance with the Declaration of Helsinki. 697, SUP-B15, and TOM-1 cell lines were obtained from DSMZ. Ph$^+$ ALL cells containing T315I mutation was kindly provided by Markus Müschen[38].

**Mice**. All mouse housing, breeding, and surgical procedures were approved by the governmental institutions of Baden-Württemberg (Regierungspräsidium Tübingen). BM cells from WT ($n = 7$, female), *Il7rα$^Δ$* ($n = 7$, female), *Il7rα$^{fl/fl}$* ($n = 7$, female), *Cxcr4$^{fl/fl}$* ($n = 3$, female) and *FoxO1$^{fl/fl}$* ($n = 3$, female) mice were collected and retrovirally transformed with either an empty pMIG vector or with a pMIG vector expressing BCR-ABL1. Unless mentioned otherwise, cells were cultured for 3–7 days in Iscove's medium (Biochrom AG) containing 10% heat-inactivated FCS (Sigma-Aldrich), 2 mM L-glutamine, 100 U/ml penicillin (Gibco), 100 U/ml streptomycin (Gibco), and 50 μM 2-mercaptoethanol. The medium was supplemented in excess with the supernatant of J558L plasmacytoma cells stably transfected with a vector encoding murine IL7. Transformed cells were selected by IL7 withdrawal and kept in optimum conditions[38]. Retroviral vectors containing either constitutively active STAT5 (STAT5-CA)[34] or an empty vector (EV) were used to transduce BCR-ABL1-transformed cells and sorted cells were used then for western blot or flow cytometry analysis. 1–2 μM 4-hydroxy Tam (Sigma-Aldrich) was used to induce deletion on plasmids expressing Tam-inducible form of Cre (Cre-ER$^{T2}$)[14]. All cells were tested and found free from mycoplasma.

**Expression assays**. Total RNA was isolated using the Direct-zol™ RNA Kit (Zymo Research) or ReliaPrep™ RNA Cell Miniprep System (Promega), and synthesis of cDNA was performed (Thermo Fisher). Quantitative real time PCR analyses were performed on ABI7900HT PCR machine (Applied Biosystems) using Quantitect assays (Qiagen) and SYBR Green (Applied Biosystems). The expression of *ABL1* and the fusion *BCR-ABL* (m-bcr; e1-a2) were measured using TaqMan Gene expression assays (Hs01104728_m1 ABL1 and Hs03024844_ft BCR-ABL, respectively) from Applied Biosystems. Relative quantification was calculated using $2^{-ΔΔCT}$ equation.

**RNA-sequencing**. BM cells were isolated from two different mice and were then kept in culture with IL7 for 7 days. Afterwards, pre-B cells were transduced with either an EV or with BCR-ABL1 retroviral vectors and kept for 48 h in +IL7 medium. Biological triplicates were prepared in three independent experiments ($n = 6$). Then, IL7 was removed from cells transduced with BCR-ABL1 for one week until cells were completely transformed. Pre-B cells transduced with EV were kept in culture with IL7 for similar culturing time points as transformed cells, then sorted for GFP. Total RNA of pre-B cells transduced with either EV or with

BCR-ABL1 was prepared using the ReliaPrep™ RNA Miniprep Kit (Promega). The total RNA library was generated using the Illumina TruSeq® stranded total RNA (Gold) kit and the multiplexed samples were sequenced on Illumina HiSeq 3000 machine to produce an average of ~100 million paired-end reads with 150 bp in length per sample. The base calling was performed by using BCL2Fastq pipeline (version: 0.3.0) and bcl2fastq (version 2.17.1.14). The broad MIT GSEA application[57] was used for GSEA. Detailed description of methods used in data analysis is provided in Supplementary Methods.

**In situ proximity ligation assay (PLA)**. For PLA experimtns[25,58], the cytokine IL7 and the chemokine CXCL12 were labeled with PLA-PLUS and PLA-MINUS probes. For PLA experiments with JAK3 or pJAK3 the corresponding antibodies were used (Cell Signaling). The PLA probes were then subjected to ligation and polymerization reactions (Sigma-Aldrich). The cells were then examined for the frequency of signals per cell under the fluorescence microscope (Leica). Pictures were taken and quantified Image J and BlobFinder software.

**In vivo transplantation of mouse leukemia cells**. Mouse pre-B cells from *Il7rα$^{fl/fl}$* or *FoxO1$^{fl/fl}$* were transformed with pMIG-BCR-ABL1 (kindly provided by W. Pear) and contained either ER$^{T2}$ or Cre-ER$^{T2}$ were labeled with retroviral firefly luciferase and were then injected intravenously into sublethally irradiated NOD-SCID mice[38]. Engraftment was monitored using luciferase bioimaging[38]. Mice were randomly allocated into each treatment group.

**Xenografts with human ALL samples**. NOD.Cg-Prkdcscid Il2rgtm1Wjl/SzJ (NSG) mice were purchased from Charles River and bred. All mouse housing, breeding, and surgical procedures were approved by the governmental animal care and use committees in Schleswig-Holstein (Ministerium für Energiewende, Landwirtschaft, Umwelt, Natur und Digitalisierung). 8–12-week-old female mice were injected intravenously with $1 × 10^6$ ALL cells from patient BM (>90% blasts)[59,60]. Animals were sacrificed upon detection of >75% leukemic blasts or clinical leukemia (loss of weight or activity, organomegaly, and hind-limb paralysis). Leukemia infiltration to spleen and BM was determined[61].

**Imatinib and antibody treatment in vivo**. NSG mice were injected with $1 × 10^6$ BCR-ABL positive ALL cells/animal. In total 40 mg/kg of imatinib (LC Laboratories) were administered orally 5 days a week. In total 1 mg/kg of anti-IL7Rα antibody (clone 40131, R&D Systems) or isotype control antibody were injected intravenously on day +1, +3, +7, +21, and then every other week. Mice were sacrificed when they showed signs of leukemia or when they had at least 75% blasts in peripheral blood. Mice were randomly allocated into each treatment group and no blinding was used.

**Flow cytometry**. Antibodies for flow cytometry (CD19, IL7R, CXCR4, FOXO1, and CD11b) were purchased from (eBioscience, BioLegend, Invitrogen, or Cell Signaling). Intracellular flow cytometry staining was performed using the Fix and Perm cell permeabilization kit (ADG). Cell viability was measured using Sytox® blue dead cell stain (Life Technologies). FACS CantoII (BD Biosciences) was used for flow cytometry, and FlowJo v.10.1 was used for data analysis. More information about the antibodies used in this study are provided in Supplementary Table 4. Detailed information for gating strategy is provided in Supplementary Fig. 14.

**Measurement of Ca$^{2+}$ flux**. A total of $1 × 10^6$ cells were loaded with Indo-1 AM (Invitrogen) and used for Ca$^{2+}$ analyses[62]. In total 100 ng/ml of CXCL12 was used for stimulation.

**Western blot**. Wet-western blotting was performed[61]. pJAK3, pJAK2, pJAK1, pSTAT5, pFOXO1, JAK1, JAK2, JAK3, STAT5, FOXO1, and GAPDH antibodies were obtained from Cell Signaling Technology. More information about the antibodies used in this study are provided in Supplementary Table 4. All originals uncropped gels are provided in Supplementary Fig. 15.

**Statistics and reproducibility**. Statistical tests are indicated in the figure legends. Results were analyzed for statistical significance with GraphPad Prism 8.3.0 software or SPSS (v 24.0.0.2). A *p* value of < 0.0500 was considered significant (*$p <$ 0.005, **$p <$ 0.001, ***$p <$ 0.001, ****$p <$ 0.001). In vitro panels are representative of at least three independent experiments, unless mentioned otherwise.

**Reporting summary**. Further information on research design is available in the Nature Research Reporting Summary linked to this article.

## Data availability

All data supporting the findings of this study are available within the article and its supplementary information files. The sequencing data that support the findings in Fig. 1,

Supplementary Figs 1 and 2, Supplementary Table 1, and Supplementary Data 1 have been deposited in NCBI's Gene Expression Omnibus[63] and are accessible through GEO Series accession number GSE150784. A reporting summary for this article is available as a Supplementary Information file.

## Code availability

The scripts used for analysis and figure generation are available at https://github.com/medhaniea/pca-and-heatmap.

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

## Acknowledgements

This work was supported by the Deutsche Krebshilfe and Deutsche Forschungsgemeinschaft (SFB1074, projects A9 and A10; SFB1279, project B03) and the ERC advanced grant to H.J. (694992). This work was also supported by the Chief Scientist Office grant to C.H. (ETM/374). H.A. is funded by the Egyptian Ministry of Higher Education (MoHE) and the German Academic Exchange Service (DAAD) within the 6th call (2014–2015; GERLS: Section 441, No.91528030). H.A. is also funded by the Department of Molecular Biology, Genetic Engineering and Biotechnology Division, National Research Centre (NRC) in Egypt. D.M.S. is supported by the Wilhelm Sander Stiftung (2016.110.1) and the Deutsche José-Carreras Leukämiestiftung (DJCLS 17R/2017). Research in the M. M. laboratory is funded by the NIH/NCI through Outstanding Investigator Award R35CA197628, R01CA157644, R01CA213138, the Norman and Sadie Lee Foundation for Pediatric Cancer and the California Institute for Regenerative Medicine (CIRM) through DISC2–10061. M.M. is a Howard Hughes Medical Institute (HHMI) Faculty Scholar. We thank Julia Lanzinger, Katrin Timm-Richert, Katrin Neumann, Annette Tietz, and Gabriele Allies for the excellent technical assistance. We thank Sebastian Wiese and the core project of SFB1074. We thank Prof. Jinyan Huang for providing RNA-Seq gene expression data for IL7R/CXCR4.

## Author contributions

H.A., A.A., A.V., M.W., A.K., and O.E. performed experiments in wildtype and BCR-ABL1-transformed murine cells. M.K. and V.S. performed PLA experiments. Z.C. and M.M performed inducible IL7R and FOXO1 deletion in vivo. A.A., L.L., and F.V. performed experiments and analyzed xenograft models. Y.Y. performed experiments. M.Y. performed RNA-Seq analysis. M.A.M. provided RNA-Seq data. E.H. provided PLA data. D.M.S., G.C., and M.S. provided ALL patient materials. A.A., E.H., D.M.S., and C.H. designed experiments and discussed the research direction. A.A. prepared all figures and wrote paper. H.J. initiated, designed, supervised research, and wrote the paper. All authors discussed the paper.

## Competing interests

The authors declare no competing interests.
