## [Peer Review File · Nature Communications]

Reviewers' comments:

Reviewer #1 (Remarks to the Author):

This study by Abdelrasoul et al explores the mechanism by which BCR-ABL transforms pre-B cells in mice and patients. They provide evidence that a complex forms including BCR-ABL, CXCR4 and IL-7Ra. This seemingly ligand-independent complex activates multiple downstream signaling pathways and is required for survival of mouse and human leukemia cells in vitro and in immunodeficient mice. This is a novel concept and can lead to therapeutic targeting of the components.

1) The model could be more clearly explained. Apparently they are proposing that IL-7Ra acts as a scaffold for these components rather than as an IL-7-responding receptor.

2) They point to development of anti-IL-7Ra as a therapeutic. The effect of their R&D Systems anti-IL-7Ra Mab in vitro and in vivo is termed "blocking". Unless there is autocrine IL-7 involved what is it blocking, formation of this complex on the cell surface? What is the evidence for that? It should not be inducing ADCC in vivo because the NSG mouse lacks NK cells, and it is a mouse IgG1 which is a poor ADCC mediator.

3) A recent publication also showed therapeutic efficacy of anti-IL-7Ra in acute lymphoblastic leukemia (Hixon et al Leukemia 2019).

Reviewer #2 (Remarks to the Author):

Overall comments:

The authors show indirect evidence to most of the claims, and several findings are not really new. The conclusion of the title is not at all supported by the experiments.

Major comments:

1.

The authors show that IL7 can act on BCR-ABL1 positive ALL leukemia cells and can rescue these leukemia cells from apoptosis upon treatment with an ABL kinase inhibitor.

This concept is not new and has been shown in various studies, including these two:

<https://www.ncbi.nlm.nih.gov/pubmed/25499760>

<https://www.ncbi.nlm.nih.gov/pubmed/23989453>

2. "BCR-ABL1 transformation requires IL7R expression":

The IL7 receptor is important for the development of lymphoid progenitors, thus it is not surprising that no transformed pre-B cells are obtained from IL7R deficient mice and that myeloid cells can be obtained.

Similarly, it is possible that IL7R is required for ALL in general, not only for BCR-ABL1 transformed cells. Thus, it will be important to demonstrate that inactivation of IL7R does not affect BCR-ABL1 independent ALL cells.

3. page 8 "these data suggest that BCR-ABL1 interaction with CXCR4 recruits this oncogene in to the proximity of IL7R associated JAK kinases..."

This conclusion is based on all indirect evidence from the experiments shown in figure 4. There is no direct evidence shown. The authors should show interaction between BCR-ABL1 and CXCR4 and/or IL7R by co-IP experiments.

4. Figure 7:

The authors show that ruxolitinib can not rescue imatinib induced inhibition of BCR-ABL1+ human ALL cells in vivo (Suppl. Fig. 7d-f). That is not surprising, since these are human leukemia cells injected in a NSG mouse: since mouse interleukin-7 is not acting on human IL7 receptor, the human leukemia cells have no stimulation of the IL7 receptor in this mouse xenograft model and there is no IL7R/JAK/STAT pathway activation. It has previously been shown that ruxolitinib in combination with dasatinib (another ABL inhibitor) is beneficial for the treatment of BCR-ABL1 ALL in a full mouse model (<https://www.ncbi.nlm.nih.gov/pubmed/25499760>).

It is then very surprising to see that targeted the IL7R protein with an antibody has effect in the xenograft model. However, the authors do not show that this is specific for BCR-ABL1+ ALL. Could it be that all ALL samples that express IL7R are sensitive to this Ab treatment ? What is the mechanism by which this Ab kills/inhibits the ALL cells ?

Reviewer #3 (Remarks to the Author):

In this study, Abdelrasoul - Alsadeq and colleagues analyzed the role of interleukin 7 receptor (IL7R) and CXCR4 in the pathogenesis of Ph-ALL and in the resistance to TKI treatment. The results suggest that Ph+ ALL requires the cooperation of IL7R with the chemokine receptor CXCR4 to recruit BCR-ABL1 and JAK kinases in close proximity thereby resulting in abnormal activation of the IL7R signaling machinery and deregulated proliferation of precursor B cells. These results have important therapeutic implications since the authors showed that treatment with anti-IL7R antibody can efficiently eliminate inhibitor-resistant Ph+ patient ALL in preclinical xenograft model, providing a new therapeutic option. However, I have several concerns about the study design and methodologies that require clarifications.

Comments

1) To better understand the molecular mechanisms regulating BCR-ABL1 induced transformation and the development of Ph+ ALL (lines 110-111, page 5) the authors performed RNA-sequencing and compared transcriptome of 6 control WT pre-B cell lines and 6 BCR-ABL-transformed pre-B cell counterparts. This experiment is crucial since defines the genes that were then functionally investigated, however clear details are not provided. First, it is not specified if a) the culturing time was the same for WT pre-B cells and BCR-ABL-transformed cells; b) the BCR-ABL-transformed cells were selected by withdrawal of IL-7 and if yes after how many days in culture; c) if transformed cells were prior RNA isolation. Second, a table with the full list of most differentially expressed genes between the two conditions needs to be provided. I wonder if gene expression data from similar datasets (pre-B cells WT and BCR-ABL transformed) are available in literature and can be used as validation of the gene expression signature here reported.

2) Among the differentially expressed genes in Figure 1a there are some with a relevant role in leukemogenesis (e.g. Bcl6 and Cdkn2a), however there is no description of their role in the manuscript. These genes should be at least discussed, and optimally their role explored. How the list of genes shown in the heatmap of Figure 1a was defined? Please, provide the list of additional GSEA enrichment pathways.

3) Figure 1 legend, line 612: did the authors mean “transduced” instead of “transfected”?

4) Figure 2b: please specify drug exposure time.

5) Figure 2c: the authors should analyze by quantitative RT-PCR the expression also of BCR-ABL to demonstrate that the upregulation of IL7R or CXCR4 expression upon BCR-ABL1 kinase inhibition affects the survival of BCR-ABL1 transformed cells. Do resistant cells develop mutations in the ABL kinase domain?

6) The data regarding the expression of IL7 receptor and CXCR4 are contradictory. In Figure 1a the expression of Il7r/Cxcr4 is significantly downregulated in BCR-ABL transformed cells compared to wild-type, however the authors by in vivo experiments show that expression of Il7r is specifically required for the initiation and the maintenance of BCR-ABL1-induced pre-B cell transformation and ALL development (lines 164-165, page 6) and that deletion of CXCR4 resulted in rapid cell death and inability of BCR-ABL1 cells to form colonies in vitro (lines 171-172). Please, clarify.

7) The authors performed a set of experiments to show that IL7R expression is likely determined by Foxo1 expression and that treatment with anti-IL7R antibody significantly delayed leukemia onset in vivo and led to a significantly expanded survival time of treated mice. These conclusions have important therapeutic implications that can eventually be extended to other leukemia subtypes with similar gene expression (e.g. Ph-like ALL). It would be noteworthy including in this study the analysis of key other ABL/kinase fusions identified in Ph-like ALL.

8) Supplementary Figure 4c: Colony formation assay for CXCR4^{fl/fl} cells transduced with BCR-ABL1 and Cre-ERT2. Cells were treated with either Et or Tam and incubated to allow colony formation for 3 weeks. I am very surprised that there are no colonies at all in the tam condition.

9) Please explore and discuss the effects in normal cells of inhibiting IL7R.

We highly appreciate the reviewers for their insightful comments and criticism, which have greatly helped us improve both the content and the presentation of our work. A point-by-point response for the reviewers' comments is presented below.

Reviewers' comments:

Reviewer #1 (Remarks to the Author):

This study by Abdelrasoul et al explores the mechanism by which BCR-ABL transforms pre-B cells in mice and patients. They provide evidence that a complex forms including BCR-ABL, CXCR4 and IL-7Ra. This seemingly ligand-independent complex activates multiple downstream signaling pathways and is required for survival of mouse and human leukemia cells in vitro and in immunodeficient mice. This is a novel concept and can lead to therapeutic targeting of the components.

We thank the reviewer for the overall positive view of our manuscript.

1) The model could be more clearly explained. Apparently they are proposing that IL-7Ra acts as a scaffold for these components rather than as an IL-7-responding receptor.

We appreciate the reviewer's suggestion. We have added additional paragraph in the discussion to better clarify our model. In addition, we have added additional experiments and analyses in our revised manuscript to further clarify the interaction between CXCR4 and IL7R with BCR-ABL (New Supplementary Figure 11).

2) They point to development of anti-IL-7Ra as a therapeutic. The effect of their R&D Systems anti-IL-7Ra Mab in vitro and in vivo is termed "blocking". Unless there is autocrine IL-7 involved what is it blocking, formation of this complex on the cell surface? What is the evidence for that? It should not be inducing ADCC in vivo because the NSG mouse lacks NK cells, and it is a mouse IgG1 which is a poor ADCC mediator. We thank the reviewer for raising this important question. We had mentioned mistakenly the word blocking once and now in the revised version we changed it into 'targeting' which is of course the more accurate description

Since this is an important point we have performed additional experiments to show that the R&D antibody which is used in our targeting experiments doesn't block IL7 binding (New Supplementary Figure11a). Next, we investigated whether the R&D antibody interferes with the IL7R-CXCR4 complex. We have performed PLA assay in SUP-B15 Ph+ ALL cells which were treated with the antibody and found a significant reduction in the association between IL7R and CXCR4 on cell surface. (New Supplementary Figure11b).

We agree with the reviewer that ADCC doesn't play a role in our NSG model due to the absence of NK cells. In addition, our data showed that complement system was also not activated in response to anti-IL7R antibody *in vitro* (data not shown). Accordingly, we have performed proteome profiler apoptosis array (R&D) and found an upregulation of several apoptotic proteins in response to IL7R antibody (Data not

shown). Therefore, we propose that IL7R antibody induces apoptosis by disrupting the IL7R/CXCR4 signaling complex.

To further confirm our findings, we have performed *in vitro* experiments (New Supplementary Figure11c) showing that treating Ph+ ALL patient derived xenograft cells with anti-IL7R antibody induced the cleavage of Caspase-8 which is a sign of apoptosis (Tummers and Green 2017.)

3) A recent publication also showed therapeutic efficacy of anti-IL-7Ra in acute lymphoblastic leukemia (Hixon et al Leukemia 2019)

As suggested by the reviewer we have discussed and cited the study of Hixon et al (Hixon et al. 2019).

Reviewer #2 (Remarks to the Author):

Overall comments:

The authors show indirect evidence to most of the claims, and several findings are not really new. The conclusion of the title is not at all supported by the experiments.

Major comments:

1.The authors show that IL7 can act on BCR-ABL1 positive ALL leukemia cells and can rescue these leukemia cells from apoptosis upon treatment with an ABL kinase inhibitor.

This concept is not new and has been shown in various studies, including these two:
<https://www.ncbi.nlm.nih.gov/pubmed/25499760>
<https://www.ncbi.nlm.nih.gov/pubmed/23989453>

We do not claim that IL7-mediated rescue of BCR-ABL positive cells upon treatment with tyrosine kinase inhibitors (TKI) is a novel finding. Our data propose an unknown mechanism for BCR-ABL1 mediated transformation. This does not only explain the mechanism of IL7-mediated rescue but also suggests that interfering with the IL7R/CXCR4 interaction leads to the death of leukemic cells independent of IL7 binding. In fact, we included experiments showing that the used anti-IL7R weakens the interaction and induces apoptotic markers without interfering with IL7 binding. Thus, IL7 treatment cannot rescue these cells. Therefore, we propose that treatment with this anti-IL7R antibody is suitable for TKI-resistant Ph+ as IL7 will not be able to rescue the cells as the signaling complex for survival is disturbed by the antibody. These findings, conclusion and treatment are novel and important for new therapeutic approaches. Moreover, our data are in agreement with the title which is also supported with the new Co-IP data showing the interaction of BCR-ABL1 with IL7R and CXCR4.

In the revised version, we have referenced and discussed both studies mentioned by the reviewer.

2. "BCR-ABL1 transformation requires IL7R expression":

The IL7 receptor is important for the development of lymphoid progenitors, thus it is not surprising that no transformed pre-B cells are obtained from IL7R deficient mice and that myeloid cells can be obtained.

We thank the reviewer for pointing out this issue. Indeed, IL7Ra deficient mice had some pro-B cells which we could efficiently transduce with BCR-ABL1 as shown in

Supplementary Figure 4. We also have data showing the presence of CD19+ population at day 1 after transformation. Nevertheless, we have added an additional reference from an independent study (Peschon et al., 1994) which also shows that IL7R deficient mice have early progenitor B cells.

Importantly, we have validated the results of this experiment using an inducible deletion model. As shown in Figure 3d-e we used Cre-ERT2 system to induce IL7R deletion in IL7R^{fl/fl} pre-B cells transformed with BCR-ABL1 and found that IL7R expression is indeed required for their survival. These data are unambiguous and in clear support of the continuous importance of IL7R for BCR-ABL1 induced transformation.

Similarly, it is possible that IL7R is required for ALL in general, not only for BCR-ABL1 transformed cells. Thus, it will be important to demonstrate that inactivation of IL7R does not affect BCR-ABL1 independent ALL cells.

We agree with the reviewer regarding an important role of IL7R in ALL regardless of the chromosomal translocation. Actually, our previous publication (Alsadeq et al., 2018) shows that IL7R is expressed in BCP-ALL regardless of the genetic entity and an increased expression levels of IL7R was associated with central nervous system (CNS) involvement and CNS-relapse. In the current study, however, we introduce a new model to investigate the molecular mechanisms of the IL7R role in the malignant transformation induced by specific oncogenes such as BCR-ABL1. We show that CXCR4 and IL7R form a complex that interacts with the fusion protein BCR-ABL1 and is important for oncogenic signaling. Importantly, we present data showing that interfering with this complex or the deletion of either IL7R or CXCR4 influences the development and survival of BCR-ABL1-positive ALL cells. This model is important to understand the mechanism of transformation and develop novel therapeutic approaches for Ph+ ALL, particularly for patients with TKI resistance.

3. page 8 "these data suggest that BCR-ABL1 interaction with CXCR4 recruits this oncogene in to the proximity of IL7R associated JAK kinases..." This conclusion is based on all indirect evidence from the experiments shown in figure. The conclusion is based on multiple findings. First, we show that IL7R and CXCR4 are localized in close proximity and that this interaction leads to increased interaction between JAK3 and CXCR4 which is then significantly reduced in the absence of IL7R (Figure 4d and Supplementary Figure 7e). Second, we show that deletion of CXCR4 reduces the phosphorylation of JAK3. Importantly, JAK3 does not usually interact with CXCR4 and the association of IL7R with CXCR4 is a likely scenario for the reduced phosphorylation of JAK3. In the revised version, we included Co-IP data to strengthen the conclusion of CXCR4 interaction with IL7R (see point# 4).

4. There is no direct evidence shown. The authors should show interaction between BCR-ABL1 and CXCR4 and/or IL7R by co-IP experiments.

We thank the reviewer for pointing out this important point. We have performed additional experiments using immunoprecipitation method to further prove our hypothesis. In particular, we have performed immunoprecipitation of Ph+ SUP-B15 ALL cells using an antibody against BCR protein. The samples were then used to detect the presence of IL7R, CXCR4 or BCR-ABL. Our data clearly show the interaction between these proteins. Data are included in Supplementary Figure 7d.

4. Figure 7:

The authors show that ruxolitinib cannot rescue imatinib induced inhibition of BCR-ABL1+ human ALL cells in vivo (Suppl. Fig. 7d-f). That is not surprising, since these are human leukemia cells injected in a NSG mouse: since mouse interleukin-7 is not acting on human IL7 receptor, the human leukemia cells have no stimulation of the IL7 receptor in this mouse xenograft model and there is no IL7R/JAK/STAT pathway activation. It has previously been shown that ruxolitinib in combination with dasatinib (another ABL inhibitor) is beneficial for the treatment of BCR-ABL1 ALL in a full mouse model (<https://www.ncbi.nlm.nih.gov/pubmed/25499760>).

We are puzzled by this comment. We are using a different model with primary ALL samples from TKI-resistant patient while the previous paper uses murine cells. We believe that this is important for the development of potential therapies. Generally, it is not unusual that different models lead to different results. However, the hypothesis of the reviewer suggests that human cells have a disadvantage as compared with murine cells because they may not be rescued by murine IL7 under TKI treatment conditions. Our data clearly show that BCR-ABL1 activates JAK kinases. Thus, one would expect that ruxolitinib blocks the BCR-ABL1 induced JAK1/2 activation.

In order to investigate this point, we have analyzed the activation of JAK2, as an indicator for JAK activation, in xenograft Ph+ALL cells (SUP-B15) isolated from NSG mice, either control mice or mice treated with ruxolitinib. As shown in the figure below, JAK2 pathway was active in these samples and was not reduced by in vivo ruxolitinib treatment (a). Furthermore, this pathway was also active in additional Ph+ ALL samples which were isolated from NSG xenograft mouse model (b). Accordingly, we exclude that the reason why ruxolitinib didn't work in our model is due to lack of activation of IL7R signaling pathway.

We are aware of the publication from Appelmann et al., 2015 and discussed their results in our manuscript. The authors in that study also showed that ruxolitinib alone had no effect both *in vitro* and *in vivo* and they see a reduction in leukemia burden only in the presence of dasatinib. In our study, however, we have used imatinib-resistant patient materials where TKI therapy would not be the most suitable approach.

It is then very surprising to see that targeted the IL7R protein with an antibody has effect in the xenograft model. However, the authors do not show that this is specific for BCR-ABL1+ ALL. Could it be that all ALL samples that express IL7R are sensitive to this Ab treatment? What is the mechanism by which this Ab kills/inhibits the ALL cells?

We thank the reviewer for asking this important question. As we discussed in point#2, our previous publication shows that IL7R is important for BCP-ALL in general, it is possible that anti-IL7R antibody treatment could be a therapeutic option for many ALL patients particularly those with risk of CNS involvement or CNS relapse or those patients who share some genetic characters, for example Ph+-like ALL. We discussed this in our revised version of the manuscript (please also refer to answer from reviewer 3, point#7)

In the revised manuscript we also provide additional data regarding the mechanism of action of the IL7R antibody (please refer to the answer of reviewer 1, point#2).

.....
Reviewer #3 (Remarks to the Author):

In this study, Abdelrasoul - Alsadeq and colleagues analyzed the role of interleukin 7 receptor (IL7R) and CXCR4 in the pathogenesis of Ph-ALL and in the resistance to TKI treatment. The results suggest that Ph+ ALL requires the cooperation of IL7R with the chemokine receptor CXCR4 to recruit BCR-ABL1 and JAK kinases in close proximity thereby resulting in abnormal activation of the IL7R signaling machinery and deregulated proliferation of precursor B cells. These results have important therapeutic implications since the authors showed that treatment with anti-IL7R antibody can efficiently eliminate inhibitor-resistant Ph+ patient ALL in preclinical xenograft model, providing a new therapeutic option. However, I have several concerns about the study design and methodologies that require clarifications.

We appreciate the overall positive feedback of the reviewer for our manuscript. In our revised manuscript we have answered all technical and conceptual concerns of the reviewer.

Comments

1) To better understand the molecular mechanisms regulating BCR-ABL1 induced transformation and the development of Ph+ ALL (lines 110-111, page 5) the authors performed RNA-sequencing and compared transcriptome of 6 control WT pre-B cell lines and 6 BCR-ABL-transformed pre-B cell counterparts. This experiment is crucial since defines the genes that were then functionally investigated, however clear details are not provided. First, it is not specified if

a) the culturing time was the same for WT pre-B cells and BCR-ABL-transformed cells; b) the BCR-ABL-transformed cells were selected by withdrawal of IL-7 and if yes after how many days in culture; c) if transformed cells were prior RNA isolation.

We thank the reviewer for this important question, and in order to clarify this issue we added further details in the method section. Yes, both WT-pre B cells and BCR-ABL1 transformed cells were cultured for similar times. BM cells were isolated from 3 different mice and then were kept in culture with IL7 for 7 days. Afterwards, pre-B cells were transduced with either EV or with BCR-ABL and kept for 48 hours in +IL7 medium. Then, IL7 was removed from cells transduced with BCR-ABL1 and cells were cultured in absence of IL7 for 1 week until cells were completely transformed as shown below. To confirm transformation, cells transduced with EV were used as a control, as shown below these cells die in the absence of IL7.

BCR-ABL1+ cells were then collected for RNA-Seq. Cells transduced with EV were cultured for similar timepoints, but always in IL7, and were sorted for GFP from which RNA was prepared directly.

Second, a table with the full list of most differentially expressed genes between the two conditions needs to be provided. I wonder if gene expression data from similar datasets (pre-B cells WT and BCR-ABL transformed) are available in literature and can be used as validation of the gene expression signature here reported.

We agree with the reviewer that comparing our data with previous work would be valuable, however, we didn't find similar experiments comparing WT and BCR-ABL1+ pre-B cell samples. In the revised version we included the full list of differentially expressed genes between the two conditions (New Supplementary Table 1) as requested by the reviewer.

2) Among the differentially expressed genes in Figure 1a there are some with a relevant role in leukemogenesis (e.g. Bcl6 and Cdkn2a), however there is no description of their role in the manuscript. These genes should be at least discussed, and optimally their role explored. How the list of genes shown in the heatmap of Figure 1a was defined? Please, provide the list of additional GSEA enrichment pathways.

We thank the reviewer for pointing this out. In the revised manuscript, we have added additional information about our analysis strategy. In short, we have added new Supplementary Table 1 showing Gene Ontology (GO) analysis for biological processes (BP) showing genes which were regulated between the control group (EV) and BCR-ABL transformed group. We also provided additional selected lists for the GOs which we thought to be of interest namely GO-0016477_Cell_Migration, GO-0007159_Leukocyte_cell-cell adhesion, GO-0050900_Leukocyte Cell Migration, GO-0046649_Lymphocyte_Activation, GO-0046651_Lymphocyte_Proliferation, and GO-0030098_Lymphocyte_Differentiation. The heat map was constructed using genes from these GO lists and which were related to IL7R signaling.

In the revised version of the manuscript, we show additional Gene Set Enrichment Analyses (GSEA). In particular, we performed GSEA on IL7R related KEGG and REACTOME genesets from the MSigDB (Broad Institute, Inc., Massachusetts Institute of Technology, and Regents of the University of California). Of the 8 genesets analyzed, 5 showed statistically significant upregulation in BCR-ABL1 as compared to control samples (False Discovery Rate (FDR) < 0.25; New Supplementary Figure 1b and New Supplementary Table 2).

We also thank the reviewer for highlighting two important genes which were also regulated in BCR-ABL transformed pre-B cells; namely *BCL6* and *CDKN2A*. Indeed, the transcription repressor *BCL6* was previously found to be regulated by *STAT5* in response to *IL7R* signaling. Accordingly, it is possible that it participates in the feedback mechanism regulating *BCR-ABL* and *IL7R*. In the revised version of the manuscript we discuss this point. However, we feel that expanding the study to include more targets could be beyond the scope of this study at the present time.

3) Figure 1 legend, line 612: did the authors mean “transduced” instead of “transfected”?

We thank the reviewer for pointing out this mistake, we have corrected the word in the revised version of the manuscript.

4) Figure 2b: please specify drug exposure time.

We thank the reviewer for pointing out the missing data. Cells were treated with 1 μ M imatinib for 15 hours, this information is now added to the figure legends.

5) Figure 2c: the authors should analyze by quantitative RT-PCR the expression also of *BCR-ABL* to demonstrate that the upregulation of *IL7R* or *CXCR4* expression upon *BCR-ABL1* kinase inhibition affects the survival of *BCR-ABL1* transformed cells. Do resistant cells develop mutations in the *ABL* kinase domain?

We appreciate very much that the reviewer has pointed out this important point. In order to properly investigate this issue we performed a new experiment, which is now shown in the new Supplementary Figure 10. In particular, 5 different WT pre-B cells transformed with *BCR-ABL1* were treated with either vehicle or imatinib (1 μ M) for 18 days to test whether we can induce imatinib-resistance (treatment schedule is shown in Supplementary Figure 10a). By day 18, we could establish imatinib-resistant cell lines. Importantly, all of imatinib-resistant cell lines expressed higher levels of *BCR-ABL1* fusion in comparison to their control counterparts. Interestingly, the two cell lines which showed the highest imatinib-resistance also showed the highest upregulation in *BCR-ABL* fusion expression levels. Nevertheless, the tested cell lines did not show mutations in the kinase domain, indicating that long-exposure to imatinib led to imatinib-resistance through upregulation of *BCR-ABL* fusion protein.

6) The data regarding the expression of *IL7* receptor and *CXCR4* are contradictory. In Figure 1a the expression of *Il7r/Cxcr4* is significantly downregulated in *BCR-ABL* transformed cells compared to wild-type, however the authors by in vivo experiments show that expression of *Il7r* is specifically required for the initiation and the maintenance of *BCR-ABL1*-induced pre-B cell transformation and ALL development (lines 164-165, page 6) and that deletion of *CXCR4* resulted in rapid cell death and inability of *BCR-ABL1* cells to form colonies in vitro (lines 171-172). Please, clarify.

We thank the reviewer for this important question. First of all, in this study we show that *BCR-ABL1* leads to downregulation of *IL7R* and *CXCR4* expression, however, transformation with *BCR-ABL1* leads to significant increase in the formation of *IL7R-CXCR4* complex when compared with untransformed WT pre-B cells. The interaction is now further confirmed by the new Co-IP experiments. We propose that *BCR-ABL1* interaction with *CXCR4* recruits this oncogene into the proximity of *IL7R*-associated *JAK* kinases thereby enabling their *BCR-ABL1* mediated activation and pre-B cell transformation. Nevertheless, according to our data *BCR-ABL1* seems to control *IL7R* expression by activating a common *STAT5*-regulated negative feedback loop. However, both *IL7R* and *CXCR4* expression is absolutely required for the survival of

BCR-ABL1 transformed pre-B cells as shown by our inducible deletion system through Cre-ERT2. We also show that targeting the IL7R in PDX pre-clinical model leads to leukemic cell death. Moreover, in the new version of the manuscript, we have performed new experiments showing that this antibody can disrupt the IL7R-CXCR4 complex and induces apoptosis (New Supplementary Figure 11). All these data indicate that IL7R expression is essential for the maintenance of BCR-ABL+ ALL cells, even though that the expression levels in transformed cells are relatively less than in untransformed pre-B cells. The data suggest that increased expression in untransformed cells is a marker for reduced IL7R signaling as strong signaling would activate STAT5 and lead to transcriptional down-regulation of IL7Ra.

7) The authors performed a set of experiments to show that IL7R expression is likely determined by Foxo1 expression and that treatment with anti-IL7R antibody significantly delayed leukemia onset in vivo and led to a significantly expanded survival time of treated mice. These conclusions have important therapeutic implications that can eventually be extended to other leukemia subtypes with similar gene expression (e.g. Ph-like ALL). It would noteworthy including in this study the analysis of key other ABL/kinase fusions identified in Ph-like ALL.

We thank the reviewer for reminding us of this important point. Indeed, patients with Ph-like ALL are characterized by a diverse array of kinase-activating alterations, including ABL fusions and JAK mutations, which make them excellent candidates for targeted therapy, similar to BCR-ABL1 ALL. Accordingly, it would be important to investigate the role of IL7R in this type of leukemia and highlight therapeutic alternatives for example through targeting the IL7R. In the revised version of our manuscript we discussed this point. However, intensive analyses should be performed in order to address this issue properly. Unfortunately, we were unable to perform these studies within the limited access to such samples.

8) Supplementary Figure 4c: Colony formation assay for CXCR4^{fl/fl} cells transduced with BCR-ABL1 and Cre-ERT2. Cells were treated with either Et or Tam and incubated to allow colony formation for 3 weeks. I am very surprised that there are no colonies at all in the tam condition.

We thank the reviewer for this comment and would like to explain as follows: one could see from Supplementary Figure 5d, the deletion of CXCR4 leads to cell death within 48 hours as indicated by the significant reduction in living cells by FACS. Accordingly, it is not surprising that transformed cells failed to form colonies within 3 weeks period. To further clarify this point, we have performed additional experiment to show that CXCR4 is absolutely required for cell survival. As shown in the figure below, deletion of CXCR4 in transformed BCR-ABL1 cells led to a rapid cell death. This effect could be reversed only by ectopic expression of survival factors, such as overexpressing BCL2.

Cxcr4^{fl/fl} - Cre-ER^{T2}

Day 5 after induction

9) Please explore and discuss the effects in normal cells of inhibiting IL7R.

In our *in vivo* experiments, treatment with anti-IL7R antibody did not result in side effects or obvious complications in mice. In the revised manuscript we further discuss the effects of targeting IL7R with an antibody.

Obviously, IL7R expression and function is critical for proper lymphopoiesis. Previous studies showed that mice deficient in IL7R had depletion in both B and T lymphocytes (von Freeden-Jeffry et al.1995). In humans, mutations in the IL7Ra result in severe combined immunodeficiency (SCID) which is associated with absence of T cells and normal numbers, nevertheless inactive, B cells (Puel et al. 1998).

Accordingly, targeting IL7Ra using specific antibodies may also affect T cells (Belarif et al. 2018) and lead to immunodeficiency in patients. We discussed these concerns and made clear that further experiments are required to clarify this issue. We believe that these experiments should be considered in an independent work which highlight therapeutic implications and challenges of IL7R antibody treatment.

Reviewers' comments:

Reviewer #1 (Remarks to the Author):

My comments were satisfactorily addressed. I recommend the publication of the paper with high enthusiasm.

Reviewer #2 (Remarks to the Author):

I like the hypothesis of this study, but, unfortunately, I do not feel that the data shown strongly support the hypothesis. In addition, there is a lot of data that is not contributing at all to the novelty of the work and to the evidence that IL7R and CXCR4 are important for BCR-ABL1 signaling in B-ALL.

Figures 1 to 3 provide data that is not very novel, not surprising and not contributing to test the central hypothesis of this work:

- Figure 1: data show that IL7R and many genes implicated in IL7R signaling are strongly downregulated in BCR-ABL1 transformed cells compared to empty vector transduced pre-B-cells. I have 2 questions about these data: (1) Do BCR-ABL1 positive B-ALL samples also show downregulation of IL7R and CXCR4 compared to other B-ALL samples ? (2) if IL7R and CXCR4 are important for BCR-ABL1 transformation, why are these then downregulated ?

- The authors conclude from this that: "Our data suggest that the signaling pathways of IL7R and CXCR4 are tightly regulated by the activity of the oncogenic kinase BCR-ABL1 and might therefore be directly involved in malignant transformation." There are many more genes deregulated by BCR-ABL1 and not all of these are involved in malignant transformation. I am not sure this conclusion makes a lot of sense since IL7R and CXCR4 seem to be strongly downregulated by BCR-ABL1.

- The authors show next that inhibition of BCR-ABL1 results in upregulation of IL7R and CXCR4, which is logic, since they first show that BCR-ABL1 suppresses these genes. I do not see why this is "interesting" it is expected.

- The authors show that conditional deletion of the IL7R gene leads to apoptosis of BCR-ABL1 transformed cells or that BCR-ABL1 transformed pre-B cells could not develop in vivo. This is interesting, but as questioned previously: is this specific for BCR-ABL1 induced transformation? I still believe that this is important to demonstrate, given the fact that the IL7R is central to B-cell development.

In my view, it would be better to integrate this part on the deletion of IL7R with the last part on the targeting of IL7R with the antibody.

Figure 4 is convincing and is the first figure demonstrating a direct link between BCR-ABL1 and CXCR4 signaling.

The part about FOXO1 is not essential and can be deleted. It disrupts the flow of the paper and it is not essential at all.

I now understand that the anti-IL7R antibody used in this study is not blocking IL7 binding, but is suggested to act by disrupting the scaffold between IL7R and CXCR4. This is now shown by limited experiments in supplemental figure 11. As this is an important part of the study, I feel that this would better be shown in a main figure and with more convincing data (more cell lines, in BCR-ABL1 transformed pre-B cells as used in other figures, in primary human B-ALL cells).

Reviewer #3 (Remarks to the Author):

The authors performed additional experiments to address all issues raised by the referees and the results from these analyses improved the manuscript. I do not have additional comments.

Reviewers' comments:

Reviewer #2 (Remarks to the Author):

I like the hypothesis of this study, but, unfortunately, I do not feel that the data shown strongly support the hypothesis. In addition, there is a lot of data that is not contributing at all to the novelty of the work and to the evidence that IL7R and CXCR4 are important for BCR-ABL1 signaling in B-ALL.

#1-Figures 1 to 3 provide data that is not very novel, not surprising and not contributing to test the central hypothesis of this work:

We respectfully disagree with the reviewer. The reviewer claims that Figures 1-3 are not very novel and that these figures do not provide data to support the main hypothesis. Importantly, the reviewer provides no specific citations for these claims. In our previous response to reviewer #1 and #2, we already discussed that there are no comparable data to draw general conclusions from multiple studies. Most importantly, We think that these data are important to understand the experimental systems and to build a rationale for experiments shown later in the manuscript.

Figure 1 shows the differential regulation of genes involved in BCR-ABL1 transformation. This is the first piece of evidence highlighting the involvement of IL7R and CXCR4 signaling in BCR-ABL1-induced transformation. Moreover, we are not aware of other publications showing RNA sequencing data for BCR-ABL1 transformed pre-B cells in comparison to untransformed pre-B cells and the reviewer also did not provide any reference in this context. In order to make this clearer we marked the two genes in the figure.

Figure 2 is essential to first show that the correlation between IL7R and CXCR4 expression, which we showed in the previous figures in a murine BCR-ABL1 model, also exists in human Ph+ ALL patients. This has never been shown and is thus novel. Also, it shows that the regulation of IL7R and CXCR4 expression in BCR-ABL1 transduced cells is strictly dependent on BCR-ABL1 kinase activity, and that only IL7 is able to rescue transformed cells from imatinib-induced death. This is also highly novel as it has never been shown in other publications and it is highly relevant from a translational point of view as this observation may have therapeutic implications. In the revised version of the manuscript we highlighted this information.

Figure 3 is also an important figure, especially if we recall comment#2 from this reviewer in the previous review: *"The IL7 receptor is important for the development of lymphoid progenitors, thus it is not surprising that no transformed pre-B cells are obtained from IL7R deficient mice and that myeloid cells can be obtained"*.

Accordingly, Figure 3 is essential to show that IL7R expression per se is absolutely required for the survival of BCR-ABL1 transformed pre-B cells using a conditional IL7R deletion model, which is the cleanest way to show this dependency.

#2- Figure 1: data show that IL7R and many genes implicated in IL7R signaling are strongly downregulated in BCR-ABL1 transformed cells compared to empty vector transduced pre-B-cells. I have 2 questions about these data:

(1) Do BCR-ABL1 positive B-ALL samples also show downregulation of IL7R and CXCR4 compared to other B-ALL samples ?

To address this point, we first investigated whether BCR-ABL1 kinase activity regulates IL7R and CXCR4 expression in human Ph⁺ ALL cells in a similar manner as in murine pre-B cells transformed with BCR-ABL1. As shown in the Supplementary Figure 4d, treating the human Ph⁺ ALL cell line with imatinib has led to increased surface expression of both IL7R and CXCR4.

Second, searching in a mixed leukemia gene expression study (Haferlach et al., 2010) using the R2 database (<http://r2.amc.nl>) showed that IL7R and CXCR4 are expressed at reduced levels in the BCR-ABL ALL (t9; 22) subgroup in comparison to other BCP-ALL entities. The data are now shown in new Supplementary Figure 3.

2) if IL7R and CXCR4 are important for BCR-ABL1 transformation, why are these then downregulated ?

Our manuscript contains a complete section (Figures 5 and 6) to explain the mechanism of how active BCR-ABL1 regulates IL7R expression through a negative feedback mechanism including STAT5 and FoxO1. In the revised version, we further clarified the mechanism and added a new figure (see New Supplementary Figure 9 and our response to comment #7).

#3- The authors conclude from this that: "Our data suggest that the signaling pathways of IL7R and CXCR4 are tightly regulated by the activity of the oncogenic kinase BCR-ABL1 and might therefore be directly involved in malignant transformation." There are many more genes deregulated by BCR-ABL1 and not all of these are involved in malignant transformation. I am not sure this conclusion makes a lot of sense since IL7R and CXCR4 seem to be strongly downregulated by BCR-ABL1.

The statement is based on data from RNA-Seq showing that signaling pathways of IL7R and CXCR4 are indeed regulated in BCR-ABL1 transformed cells. Therefore, we hypothesized that these receptors might be involved in the malignant transformation. Nevertheless, we have modified this sentence in the revised manuscript.

The reviewer argues that since BCR-ABL1 downregulates IL7R/CXCR4, then it doesn't make sense that these genes are directly involved in the malignant transformation. We think that this argument is invalid for several reasons:

-First, throughout the manuscript, we provided direct evidence for the importance of IL7R and CXCR4 in BCR-ABL1 transformed pre-B cells (For example Figure 3, 7 and Supplementary Figure 6).

-Second, we have provided a mechanism explaining how BCR-ABL1 leads to downregulation of the receptors. As shown in Figure 5, expression of BCR-ABL1 leads to an activation of STAT5, which also acts downstream IL7R. Now, activation of STAT5 inhibits FoxO1 activation which leads to downregulation of IL7R, as a negative feedback mechanism (also see response to comment #7).

-Third, if we assumed that genes which are downregulated by BCR-ABL1 are not important for the transformation, as the reviewer suggests, then one should also expect that BCL6, which is also downregulated by BCR-ABL1, is not important. Obviously, this is not the case, since the role of BCL6 has been studied before in this

context as discussed in our manuscript (Geng et al., 2015). As a further clarification, we state in the revised manuscript that several genes are regulated and that our focus in this work is IL7R/CXCR4.

#4- The authors show next that inhibition of BCR-ABL1 results in upregulation of IL7R and CXCR4, which is logic, since they first show that BCR-ABL1 suppresses these genes. I do not see why this is "interesting" it is expected.

Our data show that BCR-ABL1 upregulation leads to down regulation of several genes including IL7R, CXCR4 and CRLF2. Similarly, the inhibition of BCR-ABL1 activity through imatinib reverses this effect as shown in Figure 2c. We do find these data interesting since only IL7/IL7R activation was able to rescue transformed cells from imatinib-induced death. If we assume that all these results are "expected", as the reviewer claims, then one would expect that corresponding ligands for CXCR4 or CRLF2, SDF-1 and TSLP respectively, should also lead to rescuing the cells which is not the case (Supplementary Figure 4c). We updated the manuscript to make this point more clear.

#5- The authors show that conditional deletion of the IL7R gene leads to apoptosis of BCR-ABL1 transformed cells or that BCR-ABL1 transformed pre-B cells could not develop in vivo. This is interest, but as questioned previously: is this specific for BCR-ABL1 induced transformation? I still believe that this is important to demonstrate, given the fact that the IL7R is central to B-cell development. In my view, it would be better to integrate this part on the deletion of IL7R with the last part on the targeting of IL7R with the antibody.

We thank the reviewer for his suggestion, however, we feel that the data of IL7R deletion in the mouse model would fit better in the first part to highlight the role of IL7R in the transformation as well as to familiarize the reader with the experimental models we are using. In addition, the targeting experiments are meant to highlight the therapeutic implications for imatinib-resistant Ph+-ALL patients, where targeting with an antibody seems to be more feasible than conditional gene deletion. Considering the reviewer's comment and these arguments we decided to leave the figure as it is.

#6-Figure 4 is convincing and is the first figure demonstrating a direct link between BCR-ABL1 and CXCR4 signaling.

We appreciate that the reviewer is satisfied with the additional data which we have provided in this Figure.

#7-The part about FOXO1 is not essential and can be deleted. It disrupts the flow of the paper and it is not essential at all.

As mentioned above (comment #1/2/3). data shown in Figures 5 and 6 are essential to understand the mechanism of how BCR-ABL1 activation regulates the expression of IL7R. We propose that BCR-ABL1 controls IL7R expression by activating a common STAT5-regulated negative feedback loop. Specifically, we show that STAT5 suppresses FoxO1 transcriptional activity. Accordingly, we suggest that

overexpression of BCR-ABL1 in transformed pre-B cells results in STAT5 activation which then leads to increased FoxO1 phosphorylation and subsequent downregulation of IL7R expression (Figure 5a-c). To further confirm this, we show that conditional deletion of FoxO1 in transformed pre-B cells (system explained in Figures 6a-c) leads to reduction in IL7R expression (Figure 6d). Moreover, we show that FoxO1 deletion affects leukemia development in mice (Figure 6e-f).

In the revised manuscript, we added two references showing that CXCR4 is a direct FoxO1 transcriptional target (Dominguez-Sola et al., 2015) and (Sander et al., 2015). Due to the important information provided by FoxO1 figure we decided to leave it in the manuscript. To avoid misunderstanding, we provided further explanations in the revised version of the manuscript and provided a graphic explaining our proposed model (New Supplementary Figure 9)

#8-I now understand that the anti-IL7R antibody used in this study is not blocking IL7 binding, but is suggested to act by disrupting the scaffold between IL7R and CXCR4. This is now shown by limited experiments in supplemental figure 11. As this is an important part of the study, I feel that this would better be shown in a main figure and with more convincing data (more cell lines, in BCR-ABL1 transformed pre-B cells as used in other figures, in primary human B-ALL cells).

We are very pleased that the mechanism of action of the antibody is now more clear. As the reviewer suggested, we have performed additional experiments in Ph+-ALL cells and observed a disruption of the complex between IL7R and CXCR4 in primary ALL cells after treatment with our anti-IL7R antibody (data not shown). Nevertheless, we feel that these information are supplemental to the main message as we cannot exclude that some anti-IL7R antibodies can, at the same time, affect the binding of IL7 and disrupt the complex between IL7R and CXCR4. Comparing these activities is beyond the focus of the current manuscript.

Reviewer #3 (Remarks to the Author):

The authors made additional studies and provided clarifications to some of the concerns raised by the reviewer 2, however some of the new experiments and/or answers need further clarifications.

1) In the rebuttal letter the authors stated: "Moreover, we are not aware of other publications showing RNA sequencing data for BCR-ABL1 transformed pre-B cells in comparison to untransformed pre-B cells". RNA-seq data are available in PMID: 26321221 and this study can represent a good independent model to validate the data in Figure 1.

2) Supplementary Figure 3: from the expression data there are two groups of Ph+ ALL One with high expression of IL7/CXCR4 and one with low expression. This should be investigated and analyzed in terms of co-occurring genomic alterations. For examples, BCR-ABL1-positive ALL is characterized in 70% of cases by IKZF1 deletions. These alterations are not investigated and explored in this study. How do these alterations impact the results here obtained?

3) The data from the MILE study are from Affymetrix gene expression profile and not from RNA-seq. Since several RNA-seq data from B-ALL are publicly available, these should be analyzed for the expression of IL7 and CXCR4 and results should be described.

4) The conclusion in the Discussion paragraph is contradictory: lines 418-419 "Since IL7R expression and function is critical for proper lymphopoiesis, targeting this pathway may have effects on other normal cells." and lines 428-429 "Therefore, we propose that treatment

with anti-IL7R antibodies is a key therapeutic approach for the management of drug resistant Ph+ ALL patients." Please can you clarify how a targeting approach that can be toxic for normal cells can be a therapeutic option?

Reviewers' comments:

Reviewer #2 (Remarks to the Author):

Review:

1. In my first review I wrote:

The authors show that IL7 can act on BCR-ABL1 positive ALL leukemia cells and can rescue these leukemia cells from apoptosis upon treatment with an ABL kinase inhibitor.

This concept is not new and has been shown in various studies, including these two:

<https://www.ncbi.nlm.nih.gov/pubmed/25499760>

<https://www.ncbi.nlm.nih.gov/pubmed/23989453>

So far, my comment has not been taken into account, so I repeat my comment with some additional comments:

In the manuscript by Appelmann (pubmed/25499760) it is shown that BCR-ABL positive leukemia initiating cells (LICs) survive better to ABL kinase inhibitor treatment in the presence of IL7. Appelmann et al. show in their manuscript figure 1A a very similar figure as shown in the current manuscript in figure 2. Thus, the set up of the current experiments and conclusions of figure 2 are not very novel, it has been published by Appelmann in 'Blood' about 5 years ago (2015). It is not cited correctly in the current manuscript.

These comments are confusing and not accurate for several reasons:

-First, the reviewer claims that we have not taken his/her comments in the first revision into account. In our first response letter we have clarified that we do not claim that IL7-mediated rescue of BCR-ABL positive cells upon treatment with tyrosine kinase inhibitors (TKI) is a novel finding. Instead, we propose an unknown mechanism for BCR-ABL1 mediated transformation. This does not only explain the mechanism of IL7-mediated rescue but also suggests that interfering with the IL7R/CXCR4 interaction leads to the death of leukemic cells independent of IL7 binding. We also included experiments showing that the used anti-IL7R antibody weakens the interaction and induces apoptotic markers without interfering with IL7 binding. Thus, IL7 treatment cannot rescue these cells. Therefore, we propose that treatment with this anti-IL7R antibody is suitable for TKI-resistant ALL cells as IL7 will not be able to rescue the cells as the signaling complex for survival is disturbed by the antibody. These findings, conclusion and treatment are novel and important for new therapeutic approaches. Moreover, using ALL patient samples further underlines the differences to the Appelmann et al. study.

Furthermore, already in the first revised version we have referenced and discussed both studies mentioned by the reviewer (in our manuscript references#47 and #48, respectively).

-Second, the reviewer refers to Figure 1a from Appelmann et al., 2015 and argues that this is very similar figure to our Figure 2. Below we show the Figure which the reviewer refers to from Appelmann et al.

Figure 1. A blunted response of IL-7-treated LICs to dasatinib is reversed by ruxolitinib. (A) LICs were treated with the indicated concentrations of dasatinib, with or without 10 ng/mL of IL-7 and 100 nM ruxolitinib. Cell viability was assessed 72 hours later using a nonradioactive cell viability assay. The calculated inhibitory concentration required to arrest 50% of the cells (IC_{50}) values and results shown are taken from 3 separate experiments, each yielding triplicate determinations for each data point. Notably, the IC_{50} of dasatinib is augmented ~20-fold by IL-7 addition and is completely reversed by ruxolitinib. (B) Cells were cultured for 8 hours

Our data are not the same as shown above for many reasons:

1. The mouse cells used in this study are derived from BM cells from *Afr*^{-/-} mice backcrossed onto B6 mice expressing or lacking common gamma chain ($c\gamma$), which were then transduced with BCR-ABL. Our mice cells in Figure 2 were derived from wildtype background and transduced with BCR-ABL1. So the mouse cells are not exactly the same, as we are neither using *Afr*-deficient nor $c\gamma$ -deficient mice.

2. Importantly, the previous study considered only downstream IL7R signaling but not IL7R expression *per se* which makes a conceptual difference to our work. According to the Appelman et al. study, it is possible to some extent to transform cells from $c\gamma$ deficient mice with BCR-ABL1. This means that the expression of IL7R, which is composed of the IL7Ra chain and the $c\gamma$, is not essential for Ph+ ALL.

3. The TKI used by Appelman et al. is dasatinib whereas in our study we used imatinib. Since dasatinib also inhibits Src kinases in addition to BCR-ABL, the rescue effect of IL7 is unclear under dasatinib treatment. Importantly, imatinib is the first line treatment in Ph+ ALL.

4. The results above show that IL7 rescues murine BCR-ABL+ cells from TKI-induced cell death, an effect which is reversed in the presence of ruxolitinib. In our Figure 2b we also show that IL7 at different concentrations (range: 1.25.-5 ng/ml) can rescue cells from imatinib-induced death *in vitro*. However, we show in later figures (Supplementary Figure 10) that this is not the case *in vivo* when human Ph+ ALL cells were used in xenograft models. Accordingly, our conclusions are clearly different from Appelman et al., 2015. Most importantly, our data show that results obtained from transformed murine pre-B cells are not necessarily reflected in primary leukemia samples. Appelman et al., 2015 did not use any human ALL samples.

5. Our study included human Ph+ ALL patient data and material, which were used to investigate IL7R/CXCR4 expression. Moreover, we confirmed our concept by performing transplantation experiments with TKI-resistant Ph+ ALL patient material bearing a known BCR-ABL mutation. Such experiments were neither shown nor discussed by Appelman et al. study.

6. Figure 2 in our manuscript contains multiple additional important findings that were not shown in the previous work by Appelman et al.: i.e., overexpression of IL7R and CXCR4 in imatinib-treated BCR-ABL+ ALL cells (at both levels: protein and RNA), that the correlation between IL7R and CXCR4 expression levels is statistically significant in Ph+ ALL patient cohort and that only IL7, but not CXCL12, can rescue BCR-ABL+ cells from imatinib-induced cell death.

-Third, the reviewer claims that the publication of Appelman is not cited correctly. In the manuscript we referred to this particular reference as follows: **“Previous reports showed that combined targeting of BCR-ABL1 and JAK2 using**

dasatinib and ruxolitinib, respectively, reduced leukemia engraftment and prolonged survival⁴⁷. However, these mice eventually relapsed and died from leukemia which suggest that ruxolitinib treatment is inefficient in vivo⁴⁷.” The reviewer claims that this is incorrect, although this is the main message of the paper as one can clearly see from the title of Appelmann et al., 2015: “Janus kinase inhibition by ruxolitinib extends dasatinib- and dexamethasone-induced remissions in a mouse model of Ph+ ALL.”

-Finally, Our manuscript includes 7 main figures and 13 supplementary figures. Nevertheless, the reviewer decides to choose one out of 5 items in a single figure out of 20 figures in total to claim that the findings are not novel. The rationale behind this conclusion is unclear.

2. In my first review I also wrote a remark on the fact that the authors write “BCR-ABL1 transformation requires IL7R expression”:

The IL7 receptor is important for the development of lymphoid progenitors, thus it is not surprising that no transformed pre-B cells are obtained from IL7R deficient mice. It is possible that IL7R is required for ALL in general, not only for BCR-ABL1 transformed cells. Thus, it will be important to demonstrate that inactivation of IL7R does not affect BCR-ABL1 independent ALL cells.

The authors still write as title for figure 3 “IL7R is required for BCR-ABL1 driven leukemogenesis.” So, I repeat my question: if the authors concluded that IL7R is required for BCR-ABL1 driven leukemogenesis, they need to do this with more controls. It is also possible to show that RNA polymerase is important for BCR-ABL1 driven leukemias, but still RNA polymerase would not be a good target for therapy... The important role for IL7R in BCR-ABL1 context can only be concluded if it is show that IL7R is not needed for BCR-ABL1 independent leukemogenesis. Since IL7R is so important for B-cell development, extra care needs to be taken when drawing such specific conclusions.

We actually do not understand the example that the reviewer is stating with RNA-polymerase. In our manuscript we had the hypothesis that IL7R, CXCR4 and BCR-ABL1 act in close proximity and establish a platform that facilitates transformation. And yes, targeting RNA polymerase may not be a good target for therapy because there are no data supporting this, however, we show data supporting the notion that targeting IL7R is a good therapy and most importantly could be the most favorable option for patients who do not respond to TKI therapy. Our model is based on several experiments shown in the paper.

However, since our model become more clear in later figures in the manuscript we revised the title of figure 3 into “IL7R is required for BCP-ALL leukemogenesis”. In addition, we updated the text to avoid any misunderstanding.

3. In my first review I also wrote about figure 7 (effect of the anti-IL7R Ab in vivo):

...However, the authors do not show that this is specific for BCR-ABL1+ ALL. Could it be that all acute lymphoblastic leukemia samples that express IL7R are sensitive to this Ab treatment?

Again, this remark is not taken seriously: no additional (BCR-ABL1 negative) PDX samples have been investigated as controls.

In the revised manuscript we clearly state that anti-IL7R antibody therapy could be an option for BCP-ALL patient who are not Ph+. In addition, we also referenced a

previous publication of Alsadeq et al., 2018, which shows the efficiency of monoclonal anti-IL7R antibody in a Ph-negative PDX model.

.....
Reviewer #3 (Remarks to the Author):

The authors made additional studies and provided clarifications to some of the concerns raised by the reviewer 2, however some of the new experiments and/or answers need further clarifications.

1) In the rebuttal letter the authors stated: "Moreover, we are not aware of other publications showing RNA sequencing data for BCR-ABL1 transformed pre-B cells in comparison to untransformed pre-B cells". RNA-seq data are available in PMID: 26321221 and this study can represent a good independent model to validate the data in Figure 1.

We thank the reviewer for pointing out this important publication. However, by looking carefully into the GSE68391 dataset provided by the above mentioned study (Churchman ML, et al. Cancer Cell 2015), we could not find data comparing untransformed pre-B cells with pre-B cells transformed with BCR-ABL1. The GSE68391 contains gene expression data of BCR-ABL transformed cells which had Ikaros mutation or were treated with various drugs or vehicle.

2) Supplementary Figure 3: from the expression data there are two groups of Ph+ ALL One with high expression of IL7/CXCR4 and one with low expression. This should be investigated and analyzed in terms of co-occurring genomic alterations. For examples, BCR-ABL1-positive ALL is characterized in 70% of cases by IKZF1 deletions. These alterations are not investigated and explored in this study. How do these alterations impact the results here obtained?

We agree with the reviewer about the potential importance of co-occurring genomic mutations in Ph+ ALL including mutations in IKZF1. It was shown previously by Mullighan et al., 2008 that a common deletion involving exons 4-7 results in expression of the IK6 isoform that lacks the N-terminal DNA-binding zinc fingers, but retains the C-terminal zinc fingers responsible for dimerization. Moreover, IK6 was shown to have a dominant negative effects, partially by mis-localizing the wildtype IKZF1 from the nucleus to the cytoplasm.

To explore whether the expression of IL7R/CXCR4 in transformed BCR-ABL+ ALL is affected by IKZF1 deletion, we looked at the expression levels of IL7R and CXCR4 in the dataset accompanying an interesting publication of Churchman ML, et al. which was published in Cancer Cell 2015. The authors have reported that IL7R expression levels are significantly upregulated in IKZF1-/- BCR-ABL+ (IK6) samples as compared to BCR-ABL control cells. On the other hand, CXCR4 showed no statistical significant differential expression, albeit a subtle downregulation in the cells holding IK6.

These data indicate that IK6 deletion in Ph+ ALL may be associated with increased IL7R expression in Ph+-ALL. Similarly, it was previously shown that Ikaros negatively regulates IL7R promoter and that IK6 in ALL patients is correlated with increased IL7R expression (Ge et al., 2016). Accordingly, it is possible that ALL cells tend to upregulate IL7R expression, which is required for survival, partially through mutating the tumor suppressor Ikaros. In the revised manuscript, we have highlighted this hypothesis and provided necessary references to allow interested readers to further investigate this issue. We believe that a further analysis in patient materials should be performed to confirm this hypothesis.

3) The data from the MILE study are from Affymetrix gene expression profile and not from RNA-seq. Since several RNA-seq data from B-ALL are publicly available, these should be analyzed for the expression of IL7 and CXCR4 and results should be described.

We thank the reviewer for his/her suggestion. To further address this point, we contacted Prof. Jinyan HUANG who kindly provided us with gene expression values of IL7R/CXCR4 based on RNA-seq dataset of 1,223 BCP-ALL patients published by the study from Li et al. in 2018 (PMID: 30487223). The results show a reduced expression levels of IL7R/CXCR4 in BCR-ABL positive BCP-ALL in comparison to BCR-ABL negative BCP-ALL patients. We have included these data in the revised manuscript (New Supplementary Figure 3b).

4) The conclusion in the Discussion paragraph is contradictory: lines 418-419 “Since IL7R expression and function is critical for proper lymphopoiesis, targeting this pathway may have effects on other normal cells.” and lines 428-429 “Therefore, we propose that treatment with anti-IL7R antibodies is a key therapeutic approach for the management of drug resistant Ph+ ALL patients.” Please can you clarify how a targeting approach that can be toxic for normal cells can be a therapeutic option?

We agree with the reviewer that this is an important point to be considered. To further clarify this issue, we discussed a recent publication in Br J Clin Pharmacol (Ellis et al., 2019). The latter is a double-blind study of a single intravenous infusion of either an anti-human IL7R antibody or placebo which was carried out in 18 healthy subjects over 24 weeks. The results show that the antibody was well tolerated; there were no serious or significant adverse events. Particularly, there was neither a short- nor long-term impact on T-lymphocyte counts or subpopulations throughout the 24-week study period. In addition, no meaningful changes were observed in absolute numbers or proportions of immune cell populations or inflammatory cytokine profiles (IL-6, tumor necrosis factor- α , interferon- γ , IL-2).

In addition, recently published results of Phase 1 study show a good safety and tolerability profile for OSE-127 (a humanized monoclonal antibody with a differentiated mechanism of action as a full-antagonist of the CD127 receptor). According to the company, all pharmacokinetic and pharmacodynamic parameters were consistent and demonstrate a dose-proportionality across the several dose-levels up to 10 mg/kg (https://ose-immuno.com/en/press-releases/?ose_product=1527).

Accordingly, it seems that despite the role of IL7R for developing lymphocytes, treating ALL patients with anti-IL7R antibody would have more therapeutic benefits than adverse effects, especially if one compares the limited group of cells affected by this targeted therapy to chemotherapy or kinase inhibitors which have more broader effect. Nevertheless, we think that further investigations should be done using IL7R antibodies for therapy. In addition, it is possible that different IL7R antibodies may lead to different results. Accordingly, future experiments should also compare different ones and compare them with currently accepted therapeutic approaches in appropriate clinical trials.

REVIEWERS' COMMENTS:

Reviewer #3 (Remarks to the Author):

I do not have further comments.